# A Tad-like apparatus is required for contact-dependent prey killing in predatory social bacteria

Sofiene Seef[1†], Julien Herrou[1†], Paul de Boissier[2], Laetitia My[1], Gael Brasseur[1], Donovan Robert[1], Rikesh Jain[1,2], Romain Mercier[1], Eric Cascales[3], Bianca H Habermann[2], Tâm Mignot[1]*

[1]Aix-Marseille Université - CNRS UMR 7283, Institut de Microbiologie de la Méditerranée and Turing Center for Living Systems, Marseille, France; [2]Aix-Marseille Université - CNRS UMR 7288, Institut de Biologie du Développement de Marseille and Turing Center for Living Systems, Marseille, France; [3]Aix-Marseille Université - CNRS UMR 7255, Institut de Microbiologie de la Méditerranée, Marseille, France

**Abstract** *Myxococcus xanthus*, a soil bacterium, predates collectively using motility to invade prey colonies. Prey lysis is mostly thought to rely on secreted factors, cocktails of antibiotics and enzymes, and direct contact with *Myxococcus* cells. In this study, we show that on surfaces the coupling of A-motility and contact-dependent killing is the central predatory mechanism driving effective prey colony invasion and consumption. At the molecular level, contact-dependent killing involves a newly discovered type IV filament-like machinery (Kil) that both promotes motility arrest and prey cell plasmolysis. In this process, Kil proteins assemble at the predator-prey contact site, suggesting that they allow tight contact with prey cells for their intoxication. Kil-like systems form a new class of Tad-like machineries in predatory bacteria, suggesting a conserved function in predator-prey interactions. This study further reveals a novel cell-cell interaction function for bacterial pili-like assemblages.

*For correspondence:
tmignot@imm.cnrs.fr

[†]These authors contributed equally to this work

## Introduction

Bacterial predators have evolved strategies to consume other microbes as a nutrient source. Despite the suspected importance of predation on microbial ecology (*Mu et al., 2020*), a limited number of bacterial species are currently reported as predatory. Amongst them, obligate intracellular predators collectively known as BALOs (e.g. *Bdellovibrio bacteriovorus*) (*Mu et al., 2020*), penetrate the bacterial prey cell wall and multiply in the periplasm, escaping and killing the host bacteria (*Laloux, 2019*). Quite differently, facultative predators (meaning that they can be cultured in absence of prey if nutrient media are provided, that is *Myxococcus, Lysobacter,* and *Herpetosiphon, Mu et al., 2020*) attack their preys extracellularly, presumably by secreting antimicrobial substances and digesting the resulting products. Among these organisms and studied here, *Myxococcus xanthus*, a delta-proteobacterium, is of particular interest because it uses large-scale collective movements to attack prey bacteria in a so-called 'wolf-pack' mechanism (*Thiery and Kaimer, 2020*).

A tremendous body of work describes how *Myxococcus* cells move and respond to signals in pure culture (*Herrou and Mignot, 2020*). In contrast, mechanistic studies of the predatory cycle have been limited. Currently, it is considered that coordinated group movements allow *Myxococcus* cells to invade prey colonies and consume them via the secretion of a number of diffusible factors, extracellular enzymes, antibiotics, and outer membrane vesicles (*Thiery and Kaimer, 2020*; *Pérez et al., 2016*; *Xiao et al., 2011*). While each of these processes could contribute to predation,

evidence for their requirement is still missing (*Thiery and Kaimer, 2020*). In addition, *Myxococcus* cells have also been observed to kill prey cells upon contact, an intriguing process during which single motile *Myxococcus* cells were observed to stop and induce *E. coli* plasmolysis within a few minutes (*Zhang et al., 2020b*). *Myxococcus* pauses were more frequent with live *E. coli* cells implying prey detection but the mechanism at work and the exact relevance of prey-contact killing for predation remain unclear. Potential contact-dependent mechanisms have been described in *Myxococcus*, including Type VI secretion (*Troselj et al., 2018*) and Outer Membrane Exchange (OME, *Sah and Wall, 2020*). However, while these processes have been implicated in Kin recognition and homeostatic regulations within *Myxococcus* colonies (*Troselj et al., 2018*; *Sah and Wall, 2020*), they remain to be clearly implicated in predation. In this study, we analyzed the importance of motility and contact-dependent killing in the *Myxococcus* predation cycle.

To explore these central questions, we first developed a sufficiently resolved imaging assay where the *Myxococcus* predation cycle can be imaged stably at the single-cell level over periods of time encompassing several hours with a temporal resolution of seconds. The exact methodology underlying this technique is described in a dedicated manuscript (*Panigrahi, 2020*); briefly, the system relates predatory patterns observed at the mesoscale with single-cell resolution, obtained by zooming in and out on the same microscopy specimen (*Figure 1*). Here, we employed it to study how *Myxococcus* cells invade and grow over *Escherichia coli* prey cells during the initial invasion stage (*Figure 1*, *Video 1*).

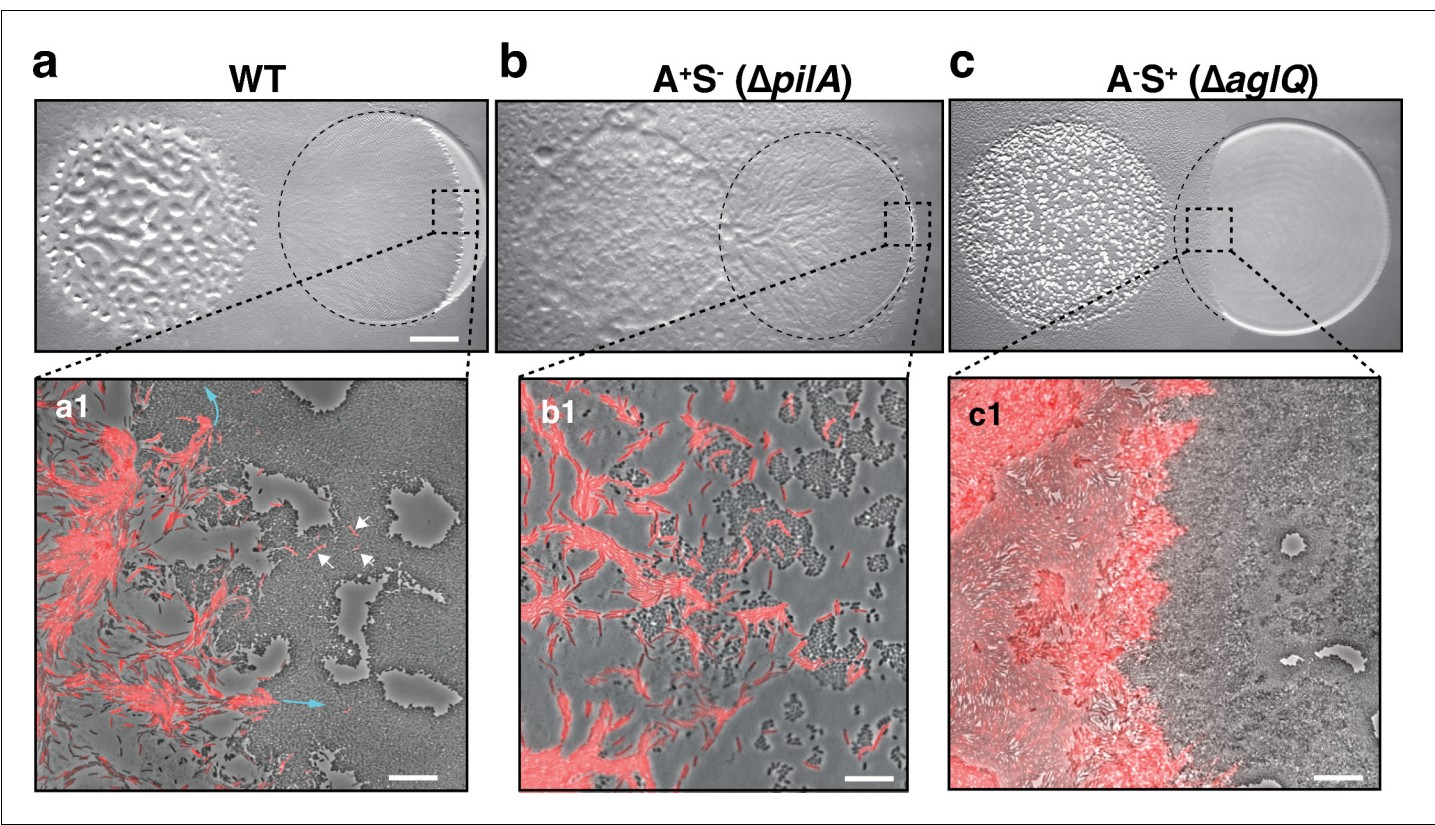

**Figure 1.** A-motility is required for invasion of prey colonies. Colony plate assays showing invasion of an *E. coli* prey colony (dotted line) 48 hr after plating by WT (**a**, *Video 1*), A⁺S⁻ (**b**) and A⁻S⁺ (**c**, *Video 2*) strains. Scale bar = 2 mm. (**a1**) Zoom of the invasion front. *Myxococcus* single cells are labeled with mCherry. Blue arrows show the movement of 'arrowhead' cell groups as they invade prey colonies. White arrows point to A-motile single cells that penetrated the prey colony. Scale bar = 10 μm. See associated *Video 1* for the full time lapse. (**b1**) Zoom of the invasion front formed by A⁺S⁻ cells. The A-motile *Myxococcus* cells can infiltrate the prey colony and kill prey cells. Scale bar = 10 μm. (**c1**) Zoom of the invasion front formed by A⁻S⁺ cells. Note that the S-motile *Myxococcus* cells come in contact with the prey colony, but in absence of A-motility, the predatory cells fail to infiltrate the colony and remain stuck at the border. Scale bar = 10 μm. See associated *Video 2* for the full time lapse.

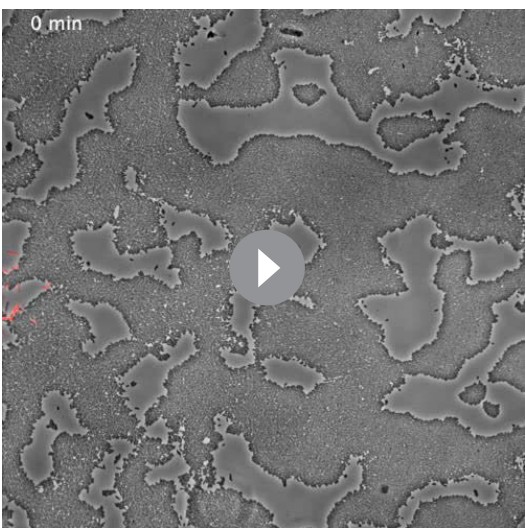
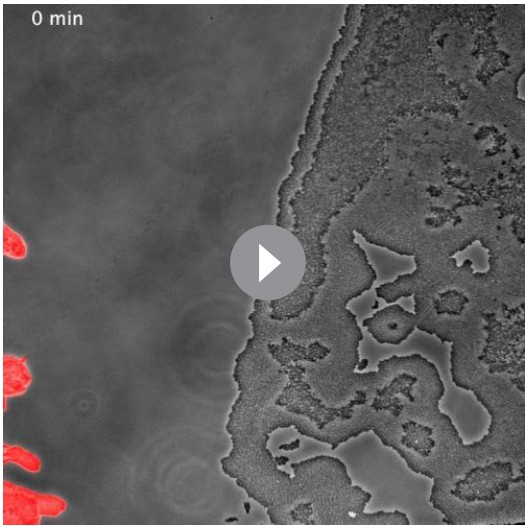

**Video 1.** Invasion of an *E. coli* colony by WT *Myxococcus* cells. This movie was taken at the interface between the two colonies during invasion. The movie is an 8x compression of an original movie that was shot for 10 hr with a frame taken every 30 s at ×40 magnification. To facilitate *Myxococcus* cells tracking, the wild-type strain was labeled with the mCherry fluorescent protein.

https://elifesciences.org/articles/72409#video1

**Video 2.** A-motility is required for prey invasion. This movie was taken at the interface between the two colonies during invasion. The movie is an 8x compression of an original movie that was shot for 10 hr with a frame taken every 30 s at ×40 magnification. To facilitate *Myxococcus* cells tracking, the A$^-$S$^+$ (Δ*aglQ*) strain was labeled with the mCherry fluorescent protein.

https://elifesciences.org/articles/72409#video2

## Results

### A-motility is required for prey colony invasion

Although the function of motility in prey invasion is generally accepted, *Myxococcus xanthus* possesses two independent motility systems and the relative contribution of each system to the invasion process is unknown. Social (S)-motility is a form of bacterial 'twitching' motility that uses so-called Type IV pili (TFP) acting at the bacterial pole (*Mercier et al., 2020*). In this process, polymerized TFPs act like 'grappling hooks' that retract and pull the cell forward. S-motility promotes the coordinated movements of *Myxococcus* cells within large cell groups due to interaction with a self-secreted extracellular matrix formed of Exo-Polysaccharide (EPS) (*Hu et al., 2016*; *Li et al., 2003*; *Islam et al., 2020*). A(Adventurous)-motility promotes the movement of *Myxococcus* single cells at the colony edges. A-motility is driven by a mobile cell-envelope motor complex (named Agl-Glt) that traffics in helical trajectories along the cell axis, driving rotational propulsion of the cell when it becomes tethered to the underlying surface at so-called bacterial Focal Adhesions (bFAs) (*Faure et al., 2016*). We tested the relative contribution of each motility system to prey invasion by comparing the relative predatory performances of WT, A$^+$S$^-$ (Δ*pilA, Sun et al., 2011*) and A$^-$S$^+$ (-Δ*aglQ, Sun et al., 2011*) strains (*Figure 1*). Interestingly, although A$^+$S$^-$ cells were defective in the late developmental steps (fruiting body formation), they were still proficient at prey invasion (*Figure 1b*). On the contrary, the A$^-$S$^+$ strain was very defective at prey colony invasion (*Figure 1c*). Zooming at the prey colony border, it was apparent that the A$^-$S$^+$ cells were able to expand and contact the prey colony, but they were unable to penetrate it efficiently, suggesting that Type IV pili on their own are not sufficient for invasion (*Figure 1c*, *Video 2*). Conversely, A-motile cells were observed to penetrate the tightly-knitted *E. coli* colony with single *Myxococcus* cells moving into the prey colony, followed by larger cell groups (*Figure 1a*). Similar motility requirements were also observed in a predatory assay where predatory and prey cells are pre-mixed before they are spotted on an agar surface (see *Figure 2* and *Figure 2—figure supplement 1*). Thus, A-motility is the main driver of prey invasion on surfaces.

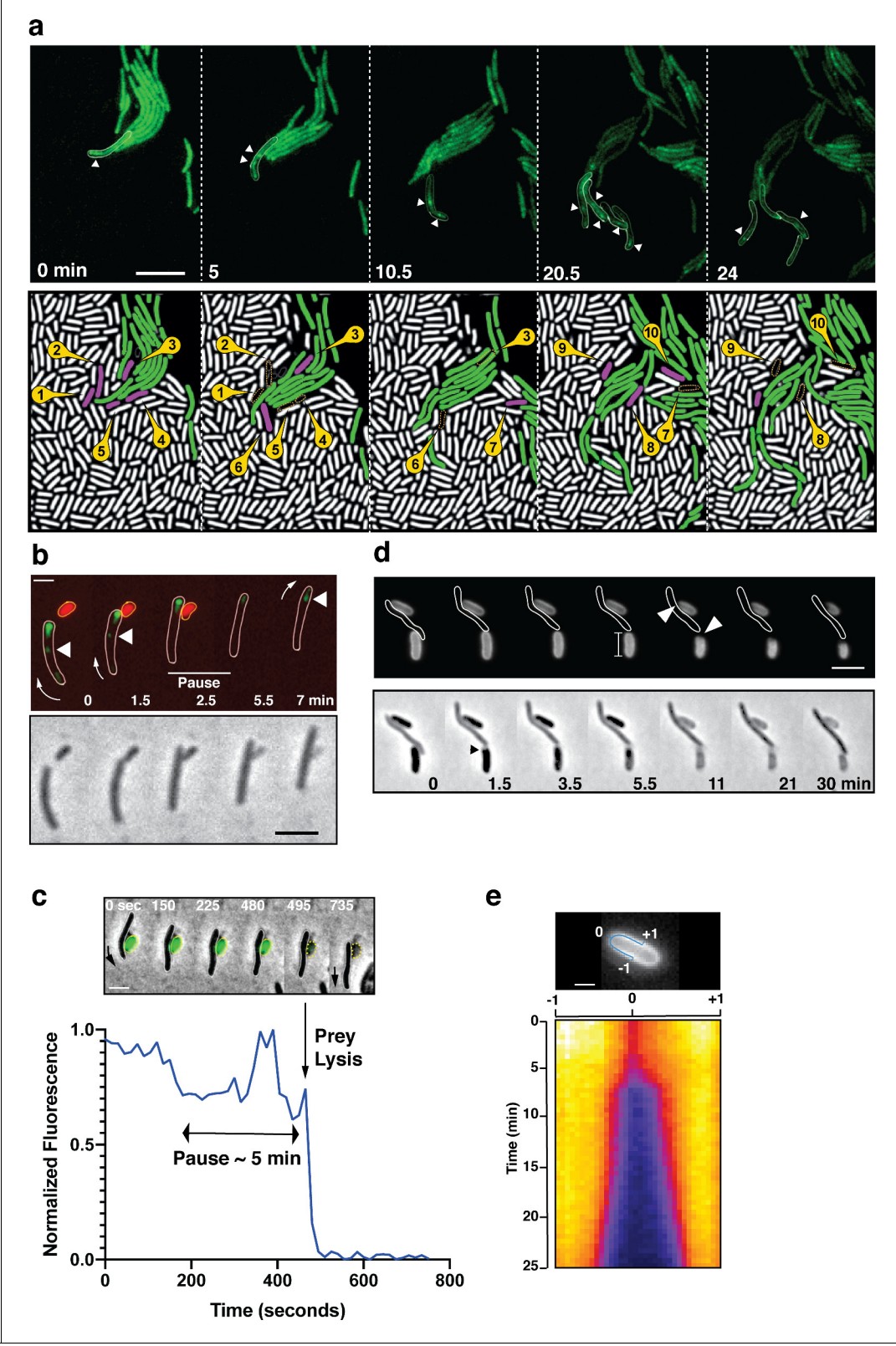

**Figure 2.** A-motile cells kill prey cells by contact. (a) Prey (*E. coli*) colony invasion by an 'arrowhead formation'. Activity of the A-motility complex is followed by monitoring *Myxococcus* cells expressing the bFA-localized AglZ-YFP protein. Upper panel: Cells within the arrowhead (outlined in white) assemble bFAs (white arrowheads). Lower panel: Semantic segmentation (see methods) of the total cell population, *E. coli* (white) and *Myxococcus* (green). The numbered and colored *E. coli* cells (magenta) are the ones that are observed to lyse as the *Myxococcus* cells penetrate the colony. See *Figure 2 continued on next page*

*Figure 2 continued*

associated *Video 3* for the full time lapse. Scale bar = 10 µm. (**b**) bFAs are disassembled when *Myxococcus* establishes lytic contacts with prey cells. Shown is an AglZ-YFP expressing *Myxococcus* cell establishing contact with an mCherry-expressing *E. coli* cell (overlay and phase contrast image). Note that the *Myxococcus* cell resumes movement and thus re-initiates bFA (white arrowheads) formation immediately after *E. coli* cell lysis. See associated *Video 4* for the full time lapse. Scale bar = 2 µm. (**c**) *Myxococcus* (outlined in white) provokes *E. coli* plasmolysis. Top: shown is a GFP-expressing *E. coli* cell lysing in contact with a *Myxococcus* cell. GFP fluorescence remains stable for 5 min after contact and becomes undetectable instantaneously, suggesting plasmolysis of the *E. coli* cell. Scale bar = 2 µm. Bottom: graphic representation of fluorescence intensity loss upon prey lysis. (**d,e**) *Myxococcus* contact provokes local degradation of the *E. coli* peptidoglycan. (**d**) *E. coli* PG was labeled covalently with the fluorescent D-amino acid TADA. Two *E. coli* cells lyse upon contact. Holes in the PG-labelling are observed at the contact sites (white arrows). Note that evidence for plasmolysis and local IM membrane contraction is visible by phase contrast for the lower *E. coli* cell (dark arrow). Scale bar = 2 µm. (**e**) Kymograph of TADA-labeling corresponding to the upper *E. coli* cell. At time 0, which corresponds to the detection of cell lysis, a hole is detected at the contact site and propagates bi-directionally from the initial site showing that the prey cell wall is degraded in time after cell death. Scale bar = 1 µm.

The online version of this article includes the following source data and figure supplement(s) for figure 2:

**Source data 1.** *E. coil* loss of fluorescence during contact-dependent lysis (*Figure 2c*).
**Figure supplement 1.** A-motility appears to be essential for predation.
**Figure supplement 2.** Contact-dependent killing by an A⁻S⁻ motility mutant (Δ*aglQ* Δ*pilA*).
**Figure supplement 3.** Contact-dependent killing by a Δ*t6ss* mutant.
**Figure supplement 4.** T6SS VipA sheath assembly in *Myxococcus* cells during predation.
**Figure supplement 5.** Prey contact-dependent lysis is not correlated to T6SS sheath contraction.
**Figure supplement 5—source data 1.** Contact-dependent lysis and VipA-GFP dynamics.
**Figure supplement 6.** Contact-dependent lysis in liquid cultures.
**Figure supplement 7.** CPRG colorimetric assay performed on motility and T6SS mutant strains.
**Figure supplement 7—source data 1.** CPRG assay.
**Figure supplement 8.** Crystal violet assay.

## Invading *Myxococcus* cells kill prey cells upon contact

To further determine how A-motile cells invade the prey colony, we shot single cell time-lapse movies of the invasion process. First, we localized a bFA marker, the AglZ protein (*Mignot et al., 2007*) fused to Neon-Green (AglZ-NG) in *Myxococcus* cells as they penetrate the prey colony. AglZ-NG binds to the cytoplasmic face of the Agl-Glt complex and has long been used as a bFA localization marker; it generally forms fixed fluorescent clusters on the ventral side of the cell that retain fixed positions in gliding cells (*Mignot et al., 2007*). As *Myxococcus* cells invaded prey colonies, they often formed 'arrow-shaped' cell groups, in which the cells within the arrow assembled focal adhesions (*Figure 2a*, *Video 3*). *E. coli* cells lysed in the vicinity of the *Myxococcus* cells, suggesting that a contact-dependent killing mechanism (as reported by *Zhang et al., 2020b*) occurs during prey colony invasion (*Figure 2a*). To observe this activity directly, we set up a *Myxococcus - E. coli* interaction microscopy assay where predator – prey interactions can be easily studied, isolated from a larger multicellular context (see Materials and methods). In this system, A-motile *Myxococcus* cells were observed to mark a pause and disassemble bFAs when contacting *E. coli* cells (*Figure 2b*, *Video 4*, further quantified below); this pause was invariably followed by the rapid death of *E. coli*, as detected by the instantaneous dispersal of a cytosolic fluorescent protein (mCherry or GFP, *Figure 2b–c*, observed in n=20 cells). This observation suggests that the killing occurs by plasmolysis, a process which is likely to be the same as that described by *Zhang et al., 2020b*. To demonstrate this, we mixed *Myxococcus* cells with *E. coli* cells in which peptidoglycan (PG) had been labeled by fluorescent D-amino Acids (TADA *Faure et al., 2016*). TADA is covalently incorporated into the PG pentapeptide backbone and it does not diffuse

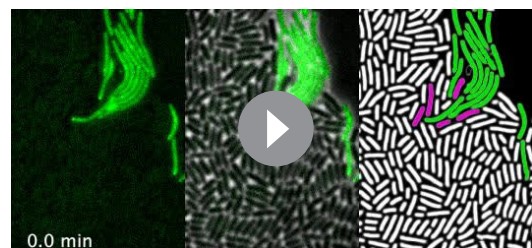

**Video 3.** Prey invasion by A-motile cells in 'arrowhead' formations. Focal adhesions and thus active A-motility complexes were detected with an AglZ-Neon green fusion. The movie contains 51 frames taken every 30 s at ×100 magnification. Shown side-by-side are fluorescence images, fluorescence overlaid with phase contrast and MiSiC segmentation (lysing *E. coli* cells are colored magenta and blue).
https://elifesciences.org/articles/72409#video3

laterally (*Kuru et al., 2012*). We first observed contraction of the *E. coli* cytosolic dense region at the pole by phase contrast (*Figure 2d*), which was followed by the appearance of a dark area in the PG TADA staining exactly at the predator-prey contact site (*Figure 2d*). It is unlikely that this dark area forms due to the new incorporation of unlabeled prey PG, because it was detected immediately upon prey cell death and propagated bi-directionally afterwards (*Figure 2d–e*). Thus, these observations suggest that, upon contact, *Myxococcus* induces degradation of the *E. coli* PG, which provokes cell lysis due to loss of turgor pressure and hyper osmotic shock (*Zhang et al., 2020b*). The bi-directional propagation of PG hydrolysis (as detected by loss of TADA signal) suggests that PG hydrolysis could be driven by the activity of PG hydrolase(s) disseminating from the predator-prey contact site.

## A predicted Tad-pilus is required for contact-dependent killing

We next aimed to identify the molecular system that underlies contact-dependent killing. Although motility appears to be essential during the predation process (*Figure 2—figure supplement 1*), at the microscopic level, direct transplantation of A⁻S⁻ (Δ*aglQ* Δ*pilA*) in *E. coli* prey colonies still exhibit contact-dependent killing (*Figure 2—figure supplement 2*), demonstrating that the killing activity is not carried by the motility complexes themselves. *Myxococcus xanthus* also expresses a functional Type VI secretion system (T6SS), which appears to act as a factor modulating population homeostasis and mediating Kin discrimination between *M. xanthus* strains (*Troselj et al., 2018*; *Vassallo et al., 2020*). A T6SS deletion strain (Δ*t6ss*) had no observable defect in contact-dependent killing of prey cells (*Figure 2—figure supplements 1* and *3*). In addition, the *Myxococcus* T6SS assembled in a prey-independent manner as observed using a functional VipA-GFP strain that marks the T6SS contractile sheath (*Brunet et al., 2013*; *Figure 2—figure supplements 4–5*), confirming that T6SS is not involved in predatory killing on surfaces.

To identify the genes (directly or indirectly) involved in the contact-dependent killing mechanism, we designed an assay where contact-dependent killing can be directly monitored in liquid cultures and observed via a simple colorimetric assay. In this system, the lysis of *E. coli* cells can be directly monitored when intracellular β-galactosidase is released in buffer containing ChloroPhenol Red-β-D-Galactopyranoside (CPRG), which acts as a substrate for the enzyme and generates a dark red hydrolysis reaction product (*Paradis-Bleau et al., 2014*). Indeed, while *Myxococcus* or *E. coli* cells incubated alone did not produce color during a 120 hr incubation, their mixing produced red color indicative of *E. coli* lysis after 24 hr (*Figure 2—figure supplement 6*). In this assay, a *t6SS* mutant was still able to lyse *E. coli* cells, demonstrating that predation is not T6SS-dependent (*Figure 2—figure supplements 6–7*). CPRG hydrolysis was not detected when *Myxococcus* and *E. coli* were separated by a semi-permeable membrane that allows diffusion of soluble molecules, showing that the assay reports contact-dependent killing (*Figure 2—figure supplement 6*). In this liquid assay, the *Myxococcus - E. coli* contacts are very distinct from contacts on solid surfaces and thus, the genetic requirements are likely quite distinct. Indeed, in the liquid CPRG assay, we observed that TFPs are essential for killing while the Agl/Glt system is dispensable (*Figure 2—figure supplement 7*). In this condition, TFPs promote a prey-induced aggregation of cells (*Figure 2—figure supplement 8*) and thus probably mediate the necessary tight contacts between *Myxococcus* and *E. coli* cells. As shown below, the killing process itself is the same in liquid as the one observed on surfaces and it is not directly mediated by the TFPs.

Given the probable indirect effect of TFPs, we next searched additional systems involved in CPRG contact-dependent killing. Using a targeted approach, we tested the effect of mutations in genome annotated cell-envelope complexes on contact-dependent killing in liquid cultures. Doing so, we identified two critical genetic regions, the MXAN_3102–3108 and the MXAN_4648–4661 regions (*Figure 3*). Functional annotations indicate that both genetic regions carry a complementary set of genes encoding proteins that assemble a so-called **T**ight **ad**herence (Tad) pilus. Bacterial Tad pili are members of the type IV filament superfamily (also including Type IV pili, a and b types, and Type II secretion systems) and extrude polymeric pilin filaments assembled via inner membrane associated motors through an OM secretin (*Denise et al., 2019*). Tad pili have been generally involved in bacterial adhesion and more recently, in contact-dependent regulation of adhesion (*Ellison et al., 2017*). The MXAN_3102–3108 cluster genes with annotated functions encode a predicted pre-pilin peptidase (CpaA and renamed KilA), a secretin homolog (CpaC/KilC) and a cytoplasmic hexameric ATPase (CpaF/KilF) (following the *Caulobacter crescentus* Tad pilus encoding *cpa* genes nomenclature, see *Figure 3a*, *Figure 3—figure supplement 1*, *Supplementary file 1*). All the other genes

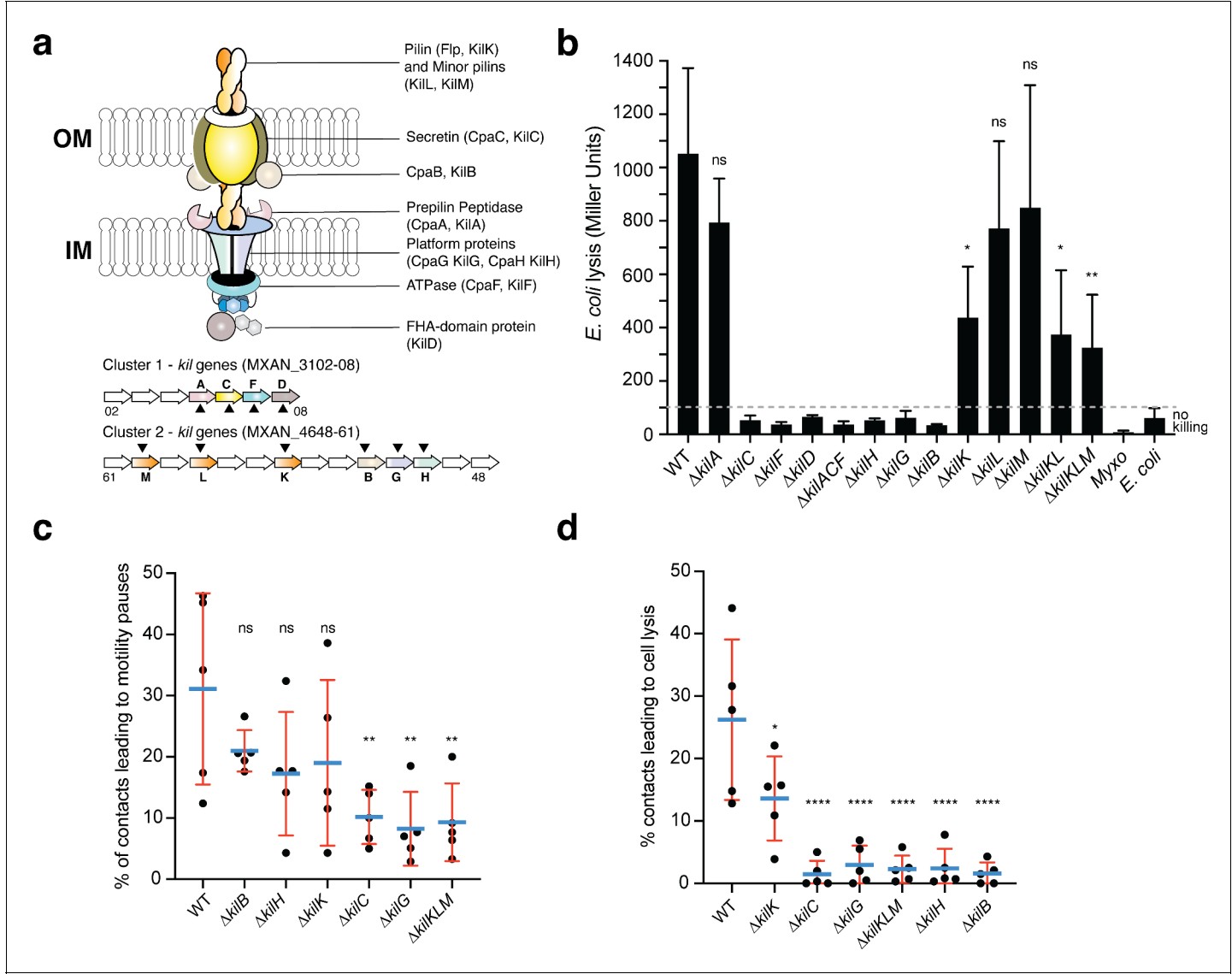

**Figure 3.** A Tad-like apparatus is required for prey recognition and contact-dependent killing. (a) Model structure of the Kil system following bioinformatics predictions. Annotated cluster 1 and cluster 2 genes are shown together with the possible localization of their protein products. Dark triangles indicate the genes that were deleted in this study. (b) *kil* mutants are impaired in *E. coli* lysis in liquid. Kinetics of CPRG-hydrolysis by the β-Galactosidase (expressed as Miller Units) observed after co-incubation of *Myxococcus* wild-type (WT) or the *kil* mutants with *E. coli* for 24 hr. *M. xanthus* and *E. coli* alone were used as negative controls. This experiment was performed independently four times. (c) The percentage of contacts with *E. coli* leading to a pause in motility was calculated for *M. xanthus* wild-type (from five independent predation movies, number of contacts observed n=807) and the *kil* mutants (number of contacts observed for Δ*kilC*: n=1780; Δ*kilH*: n=1219; Δ*kilG*: n=1141; Δ*kilB*: n=842; Δ*kilK*: n=710; Δ*kilKLM*: n=1446) (d) The percentage of contacts with *E. coli* leading to cell lysis was also estimated. In panels (b), (c), and (d), error bars represent the standard deviation of the mean. One-way ANOVA statistical analysis followed by Dunnett's posttest was performed to evaluate if the differences observed, relative to wild-type, were significant (*: p≤0.05, **: p≤0.01, ****: p≤0.0001) or not (ns: p>0.05).

The online version of this article includes the following source data and figure supplement(s) for figure 3:

**Source data 1.** CPRG assay (*Figure 3b*).

**Source data 2.** Counting percentage of contacts with a prey leading to motility pauses and prey cell lysis (*Figure 3c and d*).

**Figure supplement 1.** Bioinformatics analyses of Kil proteins.

**Figure supplement 2.** The *kil* genes are expressed during starvation.

**Figure supplement 3.** CPRG colorimetric assay.

**Figure supplement 3—source data 1.** CPRG assay.

encode proteins of unknown function, with two predicted OM lipoproteins and several proteins containing predicted ForkHead-Associated domains (FHA, *Almawi et al., 2017*, *Supplementary file 1*, see discussion). The second genetic region, MXAN_4648–4661, contains up to 14 predicted open-reading frames of which the only functionally annotated genes encode homologs of the Tad IM platform proteins (CpaG/KilG and CpaH/KilH), OM protein (CpaB/KilB), major pilin (Flp/KilK) and two pseudo-pilin subunits (KilL, M) (*Figure 3a*, *Figure 3—figure supplement 1*, *Supplementary file 1*). However, the splitting of Tad homologs in distinct genetic clusters is a unique situation (*Denise et al., 2019*) and asks whether these genes encode proteins involved in the same function.

Expression analysis suggests that the cluster 1 and cluster 2 genes are expressed together and induced in starvation conditions (*Figure 3—figure supplement 2*). We systematically deleted all the predicted Tad components in cluster 1 and 2 alone or in combination and measured the ability of each mutant to lyse *E. coli* in the CPRG colorimetric assay (*Figure 3b*). All the predicted core genes, IM platform, OM secretin and associated CpaB homolog are essential for prey lysis, with the exception of the putative pre-pilin peptidase, KilA. Deletion of the genes encoding predicted pseudo-pilins KilL and M did not affect *E. coli* killing; in these conditions, pilin fibers are only partially required because deletion of KilK, the major pilin subunit, reduces the lytic activity significantly but not fully (*Figure 3b*). Given that the genes are organized into potential operon structures, we confirmed that the CPRG-killing phenotypes of predicted cluster 1 and cluster 2 core genes were not caused by potential polar effects (*Figure 3—figure supplement 3*). In liquid, predicted core gene mutants had the same propensity as wild-type to form biofilms in presence of the prey suggesting that they act downstream in the interaction process (*Figure 2—figure supplement 8*).

We next tested whether liquid killing and contact-dependent killing on surfaces reflected the same process. For this, we analyzed selected *kil* mutants: the predicted secretin (KilC), the IM platform proteins (KilH and KilG), the OM-CpaB homolog (KilB), and the pilin and pseudopilins (KilK, L, M) in contact-dependent killing at the single cell level. Prey recognition is first revealed by the induction of a motility pause upon prey cell contact (see *Figure 2b*). This recognition was severely impaired although not fully in secretin (Δ*kilC*), IM platform protein (Δ*kilG*) and triple pilin (Δ*kilKMN*) mutants (*Figure 3c*, ~8% of the contacts led to motility pauses vs ~30% for the WT). In contrast, recognition was not impaired to significant levels in IM platform protein (Δ*kilH*), CpaB-homolog (Δ*kilB*) and pilin (Δ*kilK*) mutants (*Figure 3c*). The potential basis of this differential impact is further analyzed in the discussion. On the contrary, prey cell plasmolysis was dramatically impacted in all predicted core components (~2% of the contacts led to prey lysis vs ~26% for the WT), the only exception being the single pilin (Δ*kilK*) mutant in which prey cell lysis was reduced but still present (~13%, *Figure 3d*). Deletion of all three genes encoding pilin-like proteins were nevertheless affected in prey cell killing to levels observed in core component mutants. This is not observed to such extent in the CPRG assay, which could be explained by different cell-cell interaction requirements perhaps compensated by TFPs in liquid cultures. Given the prominent role of the pilins at the single cell level, the predicted pre-pilin peptidase KilA would have been expected to be essential. However, expression of the *kilA* gene is very low under all tested conditions (*Figure 3—figure supplement 2*). Prepilin peptidases are known to be promiscuous (*Berry and Pelicic, 2015*) and thus another peptidase (i.e. PilD, the Type IV pilus peptidase, *Friedrich et al., 2014*) could also process the Kil-associated pilins. This hypothesis could, however, not be tested because PilD appears essential for reasons that remain to be determined (*Friedrich et al., 2014*). Altogether, the data supports that the proteins from the two clusters function in starvation conditions and that they could make up a Tad-like core structure, for prey cell recognition, regulating motility in contact with prey cells, and prey killing, allowing contact-dependent plasmolysis.

## Kil proteins assemble at contact sites and mediate motility regulation and killing

We next determined if the Kil proteins indeed form a single Tad-like system in contact with prey cells. To do so, the predicted ATPase (KilF) (*Figure 3a*) was N-terminally fused to the Neon Green (NG) and expressed from the native chromosomal locus (*Figure 4a*). The corresponding fusion appeared fully functional (*Figure 4—figure supplement 1*). In absence of prey cells, NG-KilF was diffuse in the cytoplasm. Remarkably, when *Myxoccocus* cells established contact with prey cells, NG-KilF rapidly formed a fluorescent-bright cluster at the prey contact site. Cluster formation was invariably followed by a motility pause and cell lysis. Observed clusters did not localize to any specific

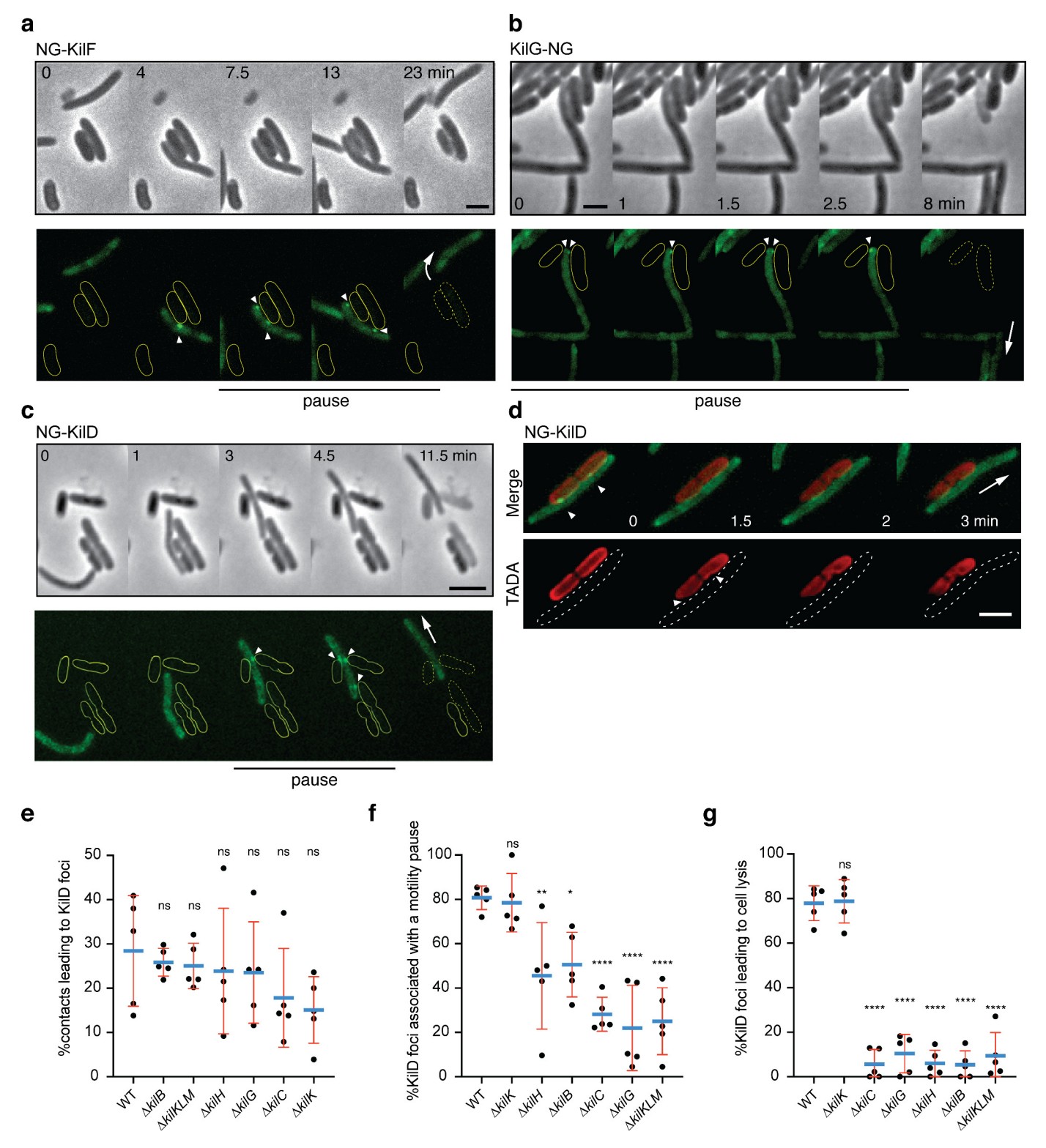

**Figure 4.** The Kil Tad-like system assembles upon contact and causes prey cell lysis. (a) NG-KilF clusters form in contact with the prey and their formation precedes cell lysis. Scale bar = 2 μm. See associated *Video 5* for the full time lapse. (b) KilG-NG forms clusters at the contact site with the prey and their formation is followed by the prey cell lysis. Scale bar = 2 μm. See associated *Video 6* for the full time lapse. (c) NG-KilD clusters only form in contact with the prey and their formation precedes cell lysis. Scale bar = 2 μm. See associated *Video 7* for the full time lapse. (d) PG-holes are formed at the cluster-assembly sites. Representative picture of TADA-labeled *E. coli* cells in the presence of NG-KilD expressing *Myxococcus xanthus*

*Figure 4 continued on next page*

*Figure 4 continued*

cells. PG holes and clusters are indicated with white arrows. Scale bar = 2 μm. (e) The percentage of contacts with *E. coli* leading to NG-KilD foci formation was calculated for *M. xanthus* wild-type (from five independent predation movies, number of contacts observed n=807) and the *kil* mutants (number of contacts observed for Δ*kilC*: n=1780; Δ*kilH*: n=1219; Δ*kilG*: n=1141; Δ*kilB*: n=842; Δ*kilK*: n=710; Δ*kilKLM*: n=1446). (f) The percentage of NG-KilD foci associated with a motility pause was also estimated for *M. xanthus* WT (from five independent predation movies, number of NG-KilD foci observed n=198) and the *kil* mutants (number of NG-KilD foci observed for Δ*kilC*: n=320; Δ*kilH*: n=270; Δ*kilG*: n=251; Δ*kilB*: n=215; Δ*kilK*: n=94; Δ*kilKLM*: n=355). (g) The percentage of NG-KilD foci leading to *E. coli* lysis was estimated as well. In panels (d), (e), and (f), error bars represent the standard deviation to the mean. One-way ANOVA statistical analysis followed by Dunnett's posttest was performed to evaluate if the differences observed, relative to wild-type, were significant (*: p≤0.05, **: p≤0.01, ****: p≤0.0001) or not (ns: p>0.05).

The online version of this article includes the following source data and figure supplement(s) for figure 4:

**Source data 1.** Counting percentage of contacts with a prey leading to NG-KilD foci formation and counting percentage of NG-KilD foci associated with motility pause and prey cell lysis (*Figure 4e, f and g*).

**Figure supplement 1.** The strains expressing Neon Green (NG) fusions of KilD or KilF have predation phenotypes similar to wild-type in a CPRG colorimetric assay.

**Figure supplement 1—source data 1.** CPRG assay.

**Figure supplement 2.** The Δ*kilG* strain expressing KilG-NG is defective in predation.

**Figure supplement 2—source data 1.** CPRG assay.

**Figure supplement 3.** Time to lysis after cluster formation.

**Figure supplement 3—source data 1.** Lysis time.

**Figure supplement 4.** Stable expression of NG-KilD in different mutant backgrounds.

**Figure supplement 4—source data 1.** Western Blot.

cellular site but they formed where *Myxococcus* cells touched the prey cells. Cluster formation was correlated to motility arrest and their dispersal coincided with motility resumption (*Figure 4a*, *Video 5*). To confirm that the NG-KilF clusters reflect assembly of a full Tad-like apparatus, we next attempted to label a component of the IM platform (KilG, gene cluster 2), expressing a KilG-NG fusion ectopically from a *pilA* promoter (*PpilA*) in a *kilG* mutant background (*Figure 4b*). The fusion was partially functional (*Figure 4—figure supplement 2*), but nevertheless KilG-NG clusters could also be observed, forming at the prey contact site immediately followed by cell lysis (*Figure 4b*, *Video 6*). These results strongly suggest that a Tad-apparatus assembled from the products of the cluster 1 and cluster 2 genes.

Additional non-core proteins are also recruited at the contact sites: downstream from *kilF* and likely co-transcribed, the MXAN_3108 gene (*kilD*, *Figure 3a* and *Figure 3—figure supplement 2*) encodes a predicted cytoplasmic multidomain protein also required for killing and thus functionally associated with the Kil apparatus (*Figure 3b*). A

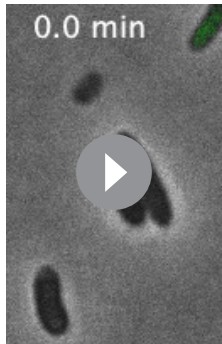

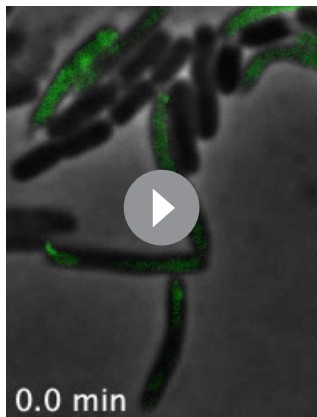

**Video 5.** NG-KilF cluster formation in contact with *E. coli* prey cells. Shown is an overlay of the fluorescence and phase contrast images of a motile *Myxococcus* cell in predatory contact with *E. coli* cells. The movie was shot at ×100 magnification objective for 30 min. Pictures were taken every 30 s.

https://elifesciences.org/articles/72409#video5

**Video 6.** KilG-NG cluster formation in contact with *E. coli* prey cells. Shown is an overlay of the fluorescence and phase contrast images of a motile *Myxococcus* cell in predatory contact with *E. coli* cells. The movie was shot at ×100 magnification objective for 9 min. Pictures were taken every 30 s.

https://elifesciences.org/articles/72409#video6

NG-KilD fusion was fully functional, also forming a fluorescent-bright cluster at a prey contact site, followed by motility arrest and prey cell lysis (*Figure 4c* and *Figure 4—figure supplement 1*, *Video 7*). As this protein is the most downstream component of the cluster 1 region, which facilitates further genetic manipulations (see below), we used it as a reporter for Kil system-associated functions. First, to confirm that prey intoxication occurs at sites where the Kil proteins are recruited, we imaged NG-KilD in the presence of *E. coli* cells labeled with TADA. As expected, PG degradation was detected at the points where the clusters are formed, showing that cluster formation correlates with contact dependent killing (*Figure 4d*). Using cluster assembly as a proxy for activation of the Kil system, we measured that killing is observed within ~2 min after assembly, a rapid effect which suggests that Kil system assembly is tightly connected to a prey cell lytic activity (*Figure 4—figure supplement 3*).

We next used NG-KilD as a proxy to monitor the function of the Kil Tad apparatus in prey recognition and killing. For this, NG-KilD was stably expressed from the native chromosomal locus in different genetic backgrounds (*Figure 4—figure supplement 4*). In WT cells, NG-KilD clusters only formed in the presence of prey cells and ~30% contacts were productive for cluster formation (*Figure 4e*). In *kil* mutants, NG-KilD clusters still formed upon prey cell contact with a minor reduction (up to twofold in Δ*kilC* and Δ*kilK*), suggesting that the Tad-like apparatus is not directly responsible for initial prey cell sensing (*Figure 4e*, *Video 8*). Nevertheless, cluster assembly was highly correlated to motility pauses (*Figure 4f*); which was impaired (up to 60%) in the *kil* mutants (except in the pilin, Δ*kilK* ) and most strongly in the Δ*kilC* (secretin), Δ*kilG* (IM platform) and triple pilin (Δ*kilKLM)* mutants (*Figure 4f*). Strikingly and contrarily to WT (and except for the major pilin (*kilK*) mutant), cluster formation was not followed by cell lysis in all *kil* mutants (or very rarely, ~4% of the time vs more than 80% in WT, *Figure 4g*). Altogether, these results indicate the Tad-like Kil system is dispensable for immediate prey recognition, but functions downstream to induce a motility pause and critically, provoke prey cell lysis.

## The kil apparatus is central for *Myxococcus* predation

We next tested the exact contribution of the *kil* genes to predation and prey consumption (*Figure 5*). This question is especially relevant because a number of mechanisms have been proposed to contribute to *Myxococcus* predation and all involve the extracellular secretion of toxic cargos (*Sah and Wall, 2020*; *Mercier et al., 2020*; *Hu et al., 2016*). In pure cultures, deletion of the *kil* genes is not linked to detectable motility and growth phenotypes, suggesting that the Tad-like Kil system mostly operates in predatory context (*Figure 5—figure supplements 1–2*). Critically, core *kil* mutants where unable to predate colonies on plate, which could be fully complemented when the corresponding *kil* genes were expressed ectopically (*Figure 5a*). When observed by time lapse, a *kil* mutant (here Δ*kilACF*) can invade a prey colony, but no prey killing is observed, showing that the prey killing phenotype is indeed due to the loss of contact-dependent killing (*Figure 5b*, *Video 9*). To measure the impact of this defect quantitatively, we developed a flow cytometry assay that directly measures the relative proportion of *Myxococcus* cells and *E. coli* cells in the prey colony across time (*Figure 5c*, see methods). In this assay, we observed that WT *Myxococcus* cells completely take over the *E. coli* population after 72 hr (*Figure 5c*). In contrast, the *E. coli* population remained fully viable when in contact with the Δ*kilACF* triple mutant, even after 72 hr (*Figure 5c*). Predatory-null phenotypes were also observed in absence of selected Tad structural components, including the secretin (KilC), the

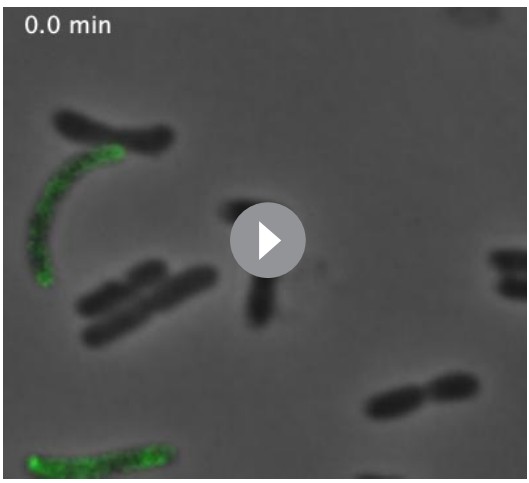

**Video 7.** NG-KilD cluster formation in contact with *E. coli* prey cells. Shown is an overlay of the fluorescence and phase contrast images of a motile *Myxococcus* cell in predatory contact with *E. coli* cells. The movie was shot at x100 magnification objective for 15 min. Pictures were taken every 30 s.
https://elifesciences.org/articles/72409#video7

ATPase (KilF) and the IM platform protein (KilH) (*Figure 5d*). A partial defect was observed in the pilin (Δ*kilK*) mutant but a triple pilin deletion mutant (Δ*kilKLM*) was, however, completely deficient (*Figure 5d*).

To further test whether Kil-dependent prey killing provides the necessary nutrient source, we directly imaged *Myxococcus* growth in prey colonies, tracking single cells over the course of 6 hr (see Materials and methods). This analysis revealed that invading *Myxococcus* cell grow actively during prey invasion. The *Myxococcus* cell cycle could be imaged directly in single cells: cell size increased linearly up to a certain length, which was followed by a motility pause and cytokinesis (*Figure 5e*, *Video 10*). The daughter cells immediately resumed growth at the same speed (*Figure 5e*). Cell size and cell age are therefore linearly correlated allowing estimation of a ~5.5 hr generation time from a compilation of traces (*Figure 5f*, n=16). When the Δ*kilACF* mutant was similarly observed, cell size tended to decrease with time and cell division was not observed (*Figure 5e–f*, n=20). Cell shortening could be a consequence of starvation, as observed for example in *Bacillus subtilis* (*Weart et al., 2007*) (although this remains to be documented in *Myxococcus*). Taken together, these results demonstrate the central function of the Kil Tad apparatus in prey killing and consumption.

## The Kil system promotes killing of phylogenetically diverse prey bacteria

*Myxococcus* is a versatile predator and can attack and digest a large number of preys (*Morgan et al., 2010*; *Müller et al., 2016*). We therefore tested if the Kil system also mediates predation by contact-dependent killing of other bacterial species. To this aim, we tested evolutionarily distant preys: diderm bacteria (*Caulobacter crescentus*, *Salmonella typhimurium* and *Pseudomonas aeruginosa*), and a monoderm bacterium (*Bacillus subtilis*). In agar plate assays, *M. xanthus* was able to lyse all tested preys, except *P. aeruginosa* (*Figure 6a–f*). When the Kil system was deleted, the predation ability of *M. xanthus* was severely diminished in all cases (*Figure 6a–f*). Consistently, *Myxococcus* assembled NG-KilD clusters in contact with *Caulobacter*, *Salmonella*, and *Bacillus* cells, which in all cases led to cell plasmolysis (*Figure 6g–i*, *Videos 11–13*). *Myxococcus* cells were however unable to form lethal clusters when mixed with *Pseudomonas* cells (tested for n=150 contacts, an example is shown in *Video 14*), suggesting that the Kil system does not assemble and thus, some bacteria can evade the prey recognition mechanism.

## The kil genes are present in other predatory delta-proteobacteria

We next explored bacterial genomes for the presence of *kil*-like genes. Phylogenetic analysis indicates that the ATPase (KilF), IM platform proteins (KilH and KilG) and CpaB protein (KilB) share similar evolutionary trajectories, allowing the construction of a well-supported phylogenetic tree based on a supermatrix (*Figure 7*, see Materials and methods). This analysis reveals that Kil-like systems are indeed related to Tad systems (i.e. Tad systems from alpha-proteobacteria, *Figure 7*) but their distribution appears to be restricted to the delta-proteobacteria, specifically in *Myxococcales*, in *Bdellovibrionales* and in the recently discovered *Bradymonadales*. Remarkably, these bacteria are all predatory and thus, given that their predicted Kil machineries are very similar to the *Myxococcus* Kil system, they could perform similar functions (*Figure 7*, *Supplementary file 2*). This is possible because *Bradymonadales* are also thought to predate by surface motility and extracellular prey attack (*Mu et al., 2020*). In addition, while at first glance it may seem that *Bdellovibrio* species have a distinct predatory cycle, penetrating the prey cell to actively replicate in their periplasmic space (*Laloux, 2019*); this cycle involves a number of processes that are similar to Myxobacteria: *Bdellovibrio* cells also attack prey cells using gliding motility (*Lambert et al., 2011*) and attach to them using Type IV pili and a number of common regulatory proteins (*Milner et al., 2014*). Prey cell penetration follows from the ability of the predatory cell to drill a hole into the prey PG at the attachment site (*Kuru et al., 2017*). Although this remains to be proven directly, genetic evidence has shown that the *Bdellovibrio* Kil homologs are important for prey invasion and attachment (*Avidan et al., 2017*; *Duncan et al., 2019*).

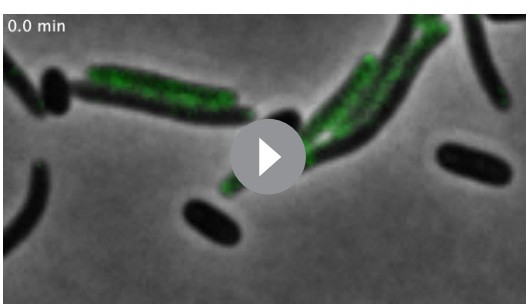

**Video 8.** NG-KilD clusters form in a Δ*kilC* mutant but no motility pauses and prey cell lysis can be observed. Shown is an overlay of the fluorescence and phase contrast images of a motile *Myxococcus* cell in predatory contact *E. coli* cells. The movie was shot at ×100 magnification objective for 5.5 min. Pictures were taken every 30 s.
https://elifesciences.org/articles/72409#video8

## Discussion

Prior to this work, *Myxococcus* predation was thought to involve motility, contact-dependent killing (*Zhang et al., 2020b*), secreted proteins, Outer Membrane Vesicles (OMVs) and antibiotics (i.e. Myxovirescin and Myxoprincomide) to kill and digest preys extracellularly (*Thiery and Kaimer, 2020*; *Pérez et al., 2016*). While a contribution of these processes is not to be ruled out, they are most likely involved in prey cell digestion (for example by degradative enzymes) rather than killing (*Thiery and Kaimer, 2020*), and we show here that contact-dependent killing is the major prey killing mechanism. In *Myxococcus*, contact-dependent killing can be mediated by several processes, now including T6SS, OME (Outer Membrane Exchange) and Kil. We exclude a function for the T6SS, for which a role in *Myxococcus* interspecies interactions has yet to be demonstrated. Rather, it appears that together with OME, Type-VI secretion controls a phenomenon called social compatibility, in which the exchange of toxins between *Myxococcus* cells prevents immune cells from mixing with non-immune cells (*Vassallo et al., 2020*). We have not tested a possible function of OME in prey killing because OME allows transfer of OM protein and lipids between *Myxococcus* cells when contact is established between identical outer membrane receptors, TraA (*Sah and Wall, 2020*). OME is therefore a process that only occurs between Myxococcus strains to mediate social compatibility when SitA lipoprotein toxins are delivered to non-immune TraA-carrying *Myxococcus* target cells (*Vassallo et al., 2017*).

The Kil system is both required for contact-dependent killing in liquid and on surfaces. Remarkably, proteins belonging to each motility systems show distinct requirements in liquid or on solid media. In liquid, Type-IV pili mediate prey-induced biofilm formation, which likely brings *Myxococcus* in close contact with the prey cells. This intriguing process likely requires EPS (since *pilA* mutants also lack EPS, *Black et al., 2006*), which deserves further exploration. On surfaces, likely a more biologically relevant context, contact-dependent killing is coupled to A-motility to penetrate prey colonies and interact with individual prey cells. The prey recognition mechanism is especially intriguing because dynamic assembly of a Tad-like system at the prey contact site is a novel observation; in general, these machineries and other Type-IV filamentous systems (*Denise et al., 2019*), such as TFPs tend to assemble at fixed cellular sites, often a cell pole (*Mercier et al., 2020*; *Ellison et al., 2017*; *Koch et al., 2021*). NG-KilD clusters do not require a functional Tad-like system to form in contact with prey cells, suggesting that prey contact induces Tad assembly via an upstream signaling cascade. Such sensory system could be encoded within the clusters 1 and 2, which contain a large number of conserved genes with unknown predicted functions (up to 11 proteins of unknown functions just considering cluster 1 and 2, *Figure 3a*, *Supplementary file 1*). In particular, the large number of predicted proteins with FHA type domains (*Almawi et al., 2017*; *Supplementary file 1*) suggests a function in a potential signaling cascade. In *Pseudomonas aeruginosa*, FHA domain-proteins act downstream from a phosphorylation cascade triggered by contact, allowing *Pseudomonas* to fire its T6SS upon contact (*Basler et al., 2013*). This mechanism is triggered by general perturbation of the *Pseudomonas* membrane (*Ho et al., 2013*), which could also be the case for the Kil system. Kil assembly is provoked both by monoderm and diderm bacteria, which suggests that prey-specific determinants are unlikely. Recognition is nevertheless non-universal and does not occur in contact with *Pseudomonas* or *Myxococcus* itself. Therefore, evasion mechanisms must exist, perhaps in the form of genetic determinants that shield cells from recognition.

The Kil Tad-like system itself is required to pause A-motility and for prey cell killing. Motility regulation could be indirect because differential effects are observed depending on *kil* gene deletions (*Figures 3* and *4*), suggesting that assembly of a functional Tad apparatus is not strictly required for

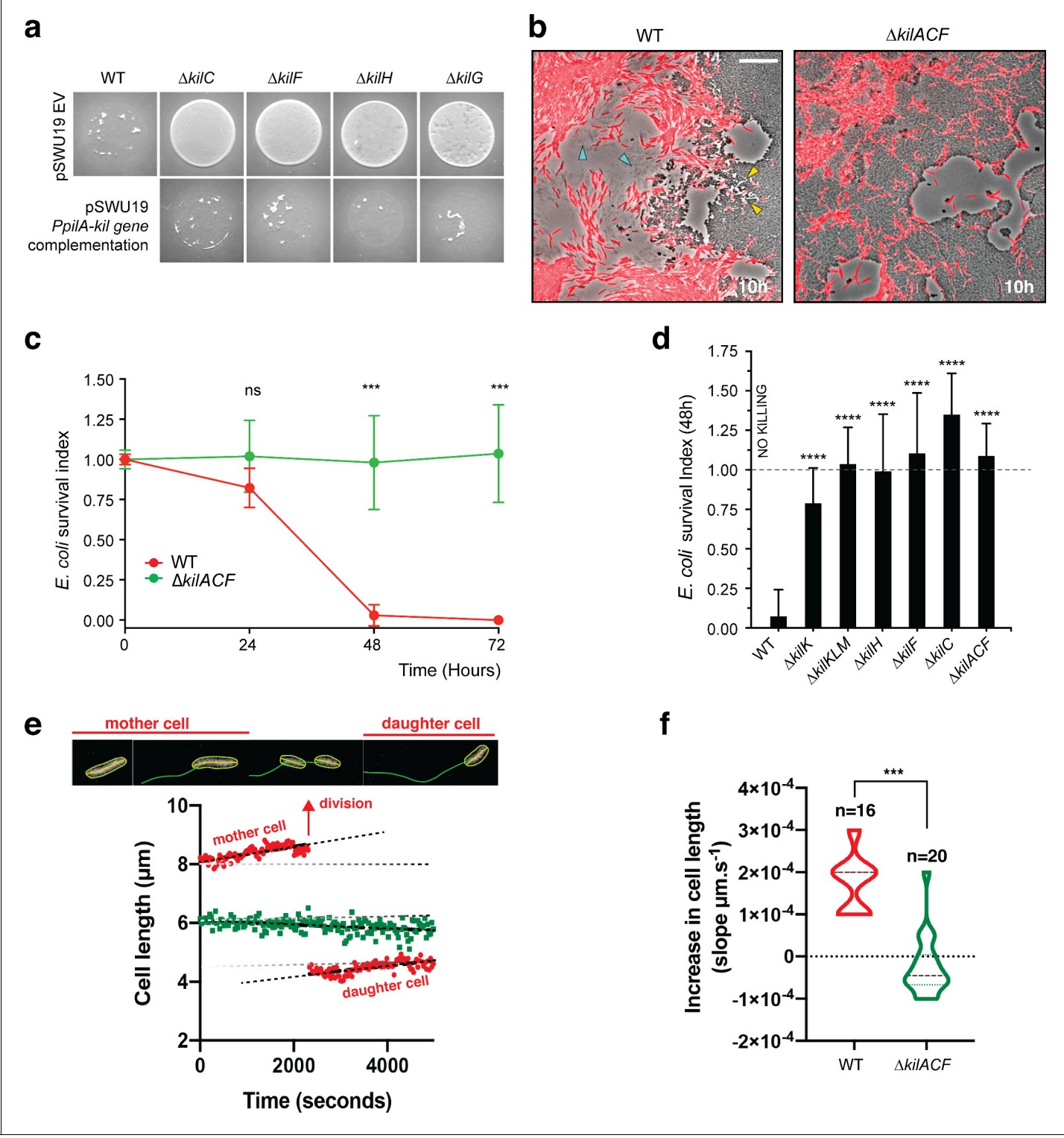

**Figure 5.** The *kil* genes are required for *M.xanthus* nutrition over prey cells. (**a**) The Kil system is essential for predation. Core deletion mutants in Tad-like genes, Δ*kilC* (Secretin), Δ*kilF* (ATPase), Δ*kilH* (IM platform), and Δ*kilG* (IM platform) were mixed with *E. coli* and spotted on CF agar plates (+ 0.07% glucose). After 24 hr of incubation, the mutants carrying the empty vector (EV) pSWU19 were strongly deficient in predation. The same *kil* mutants ectopically expressing the different *kil* genes under the control of the *pilA* promoter presented a restored predation phenotype similar to the WT-EV control. (**b**) A *kil* mutant can invade but cannot lyse *E. coli* prey colonies. mCherry-labeled WT and triple *kilACF* mutant are shown for comparison. Note that invading WT cells form corridors (yellow arrowheads) in the prey colony and ghost *E. coli* cells as well as cell debris (blue arrowheads) are left

*Figure 5 continued on next page*

*Figure 5 continued*

behind the infiltrating *Myxococcus* cells. In contrast, while the Δ*kilACF* penetrates the prey colony, corridors and prey ghost cells are not observed. Scale bar = 10 μm. See corresponding **Video 9** for the full time lapse. (**c**) The *kil* genes are essential for prey killing. *E. coli* mCherry cells were measured by flow cytometry at time 0, 24, 48, and 72 hr after the onset of predation. The *E. coli* survival index was calculated by dividing the percentage of '*E. coli* events' at t=24, 48, or 72 hr by the percentage of '*E. coli* events' at the beginning of the experiment (t=0). This experiment was independently performed three times (n=6 per strain and time point). For each sample, 500,000 events were analyzed. Each data point indicates the mean ± the standard deviation. For each time point, unpaired t-test (with Welch's correction) statistical analysis was performed to evaluate if the differences observed, relative to wild-type, were significant (***: p≤0.001) or not (ns: p>0.05). (**d**) *E. coli* survival in presence of the different *kil* mutant strains at 48 hr. *E. coli* mCherry cells were measured by flow cytometry at time 0 and 48 hr after predation. This experiment was independently performed three times (n=9 per strain and time point). Events were counted as in (c) and each data point indicates the mean ± the standard deviation. One-way ANOVA statistical analysis followed by Dunnett's posttest was performed to evaluate if the differences observed, relative to wild-type, were significant (****: p≤0.0001). (**e, f**) The *kil* genes are essential for *Myxococcus* growth on prey. (**e**) Cell growth during invasion. Cell length is a function of cell age during invasion and can be monitored over time in WT cells (in red). In contrast, cell length tends to decrease in a Δ*kilACF* mutant (in green) showing that they are not growing in presence of prey. See associated **Video 10** for the full time lapse. (**f**) Quantification of cell growth in WT and in Δ*kilACF* mutant backgrounds. Each individual cell was tracked for 5 hr in two biological replicates for each strain. Violin plot of the growth distributions (shown as the cell size increase slopes) are shown. Statistics: Student t-test, ***: p<0.001.

The online version of this article includes the following source data and figure supplement(s) for figure 5:

**Source data 1.** Flow cytometry (*Figure 5c and d*).
**Source data 2.** *M. xanthus* growth during prey colony invasion (*Figure 5e*).
**Source data 3.** Increase in *M. xanthus* cell length during predation (*Figure 5f*).
**Figure supplement 1.** Growth and motility of WT and Δ*kilACF* strains on agar supporting both A- and S-motility (1.5%) and S-motility only (0.5%).
**Figure supplement 2.** Growth curves of WT and Δ*kilACF* in CYE-rich medium.
**Figure supplement 2—source data 1.** Growth curves.

regulation. In contrast, prey killing requires a functional Tad apparatus. In particular, the pilin proteins are required during prey invasion but they are dispensable (partially) in liquid cultures showing that they do not promote toxicity. In liquid, direct contacts may be enforced by TFPs in the biofilm, perhaps rendering the Tad pilins partially redundant. Tad pilin function nevertheless becomes essential on surfaces when A-motility is active. How the pilins organize to form polymers and whether they do, remains to be determined; the lack of the major pilin (KilK) is compensated by the remaining pseudo-pilins KilL and M, which is somewhat surprising given that pseudopilins are generally considered to prime assembly of major pilin polymers (*Denise et al., 2019*). It is currently unclear if the Kil system is also a toxin-secretion device; for example, if it also functions as a Type II secretion system. Alternatively, the Kil complex might recruit a toxin delivery system at the prey contact site. This latter hypothesis is in fact suggested by the remaining low (but still detectable) contact-dependent toxicity of the *kil* mutants (*Figures 3* and *4*). Given that *Myxococcus* induces prey PG degradation locally, we hypothesize that a secreted cell wall hydrolase becomes active at the prey contact site. This is not unprecedented: *Bdellovibrio* cells secrete a sophisticated set of PG modifying enzymes, D,D-endopeptidases (*Lerner et al., 2012*), L,D trans-peptidases (*Kuru et al., 2017*) and Lysozyme-like enzymes (*Harding et al., 2020*) to penetrate prey cells, carve them into bdelloplasts and escape. In *Myxococcus*, deleting potential D,D-endopeptidases (*Zhang et al., 2020a*) (Δ*dacB*) did not affect predation (*Figure 7—figure supplement 1*) which might not be surprising given

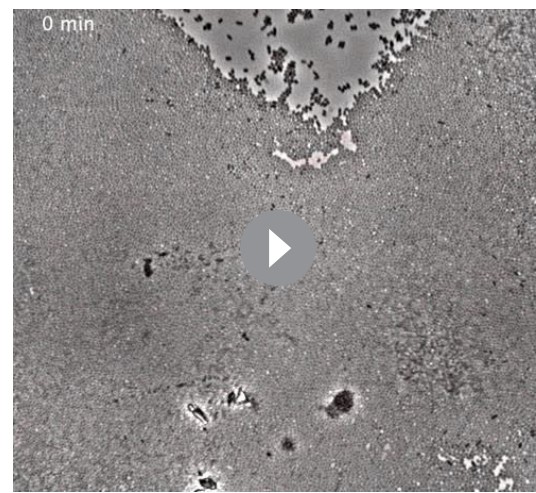

**Video 9.** A Δ*kilACF* still invades but does not kill *E. coli* prey cells. This movie was taken at the interface between the two colonies during invasion. The movie is a 4x compression of an original movie that was shot for 4.5 hr with a frame taken every 30 s at ×40 magnification. To facilitate tracking of *Myxococcus* Δ*kilACF*, cells are labeled with the mCherry fluorescent protein.
https://elifesciences.org/articles/72409#video9

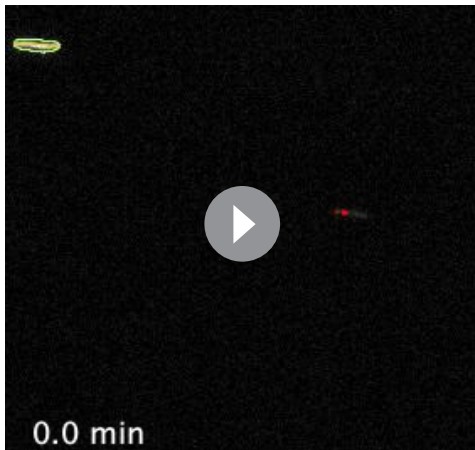

**Video 10.** Predatory cells division and tracking during invasion of prey colony. To follow cell growth and division at the single cell level during prey invasion, WT cells were mixed with a WT strain expressing the mCherry at a 50:1 ratio and imaged every 30 s at ×40 magnification for up to 10 hr within non-labeled prey colonies. Cell growth was measured by fitting cell contours to medial axis model followed by tracking under Microbe-J. Real time of the track for the example cell: 95 min.
https://elifesciences.org/articles/72409#video10

that *Myxococcus* simply lyses its preys while *Bdellovibrio* needs to penetrate them while avoiding their lysis to support its intracellular cycle. The *Myxococcus* toxin remains to be discovered, bearing in mind, that similar to synergistic toxic T6SS effectors (*LaCourse et al., 2018*), several toxic effectors could be injected, perhaps explaining how *Myxococcus* is able to kill both monoderm and diderm preys.

The Myxobacteria are potential keystone taxa in the soil microbial food web (*Petters et al., 2021*), meaning that Kil-dependent mechanisms could have a major impact in shaping soil ecosystems. While the Kil proteins are most similar to proteins from Tad systems, there are a number of key differences that suggest profound diversification: (i), the Kil system involves a single ATPase and other Tad proteins such as assembly proteins TadG, RcpB, and pilotin TadD are missing *Denise et al., 2019*; (ii), several Kil proteins have unique signatures, the large number of associated genes of unknown function; in particular, the over-representation of associated FHA domain proteins, including the central hexameric ATPase KilF itself fused to an N-terminal FHA domain. The KilC secretin is also uniquely short and lacks the N0 domain, canonically found in secretin proteins (*Tosi et al., 2014*), which could be linked to increased propensity for dynamic recruitment at prey contact sites. Future studies of the Kil machinery could therefore reveal how the contact-dependent properties of Tad pili were adapted to prey cell interaction and intoxication, likely a key evolutionary process in predatory bacteria.

## Materials and methods

### Bacterial strains, growth conditions, motility plates, genetic constructs, and western blotting

See *Supplementary files 3*, *4* and *5* corresponding to Table 3: strains, Table 4: plasmids and Table 5: primers. *E. coli* cells were grown under standard laboratory conditions in Luria-Bertani (LB) broth supplemented with antibiotics, if necessary. *M. xanthus* strains were grown at 32°C in CYE (Casitone Yeast Extract) rich media as previously described (*Zhang et al., 2020a*). *S. enterica* Typhimurium, *B. subtilis,* and *P. aeruginosa* were grown overnight at 37°C in LB. *C. crescentus* strain NA1000 was grown overnight at 30°C in liquid PYE (Peptone Yeast Extract). Motility plate assays were conducted as previously described on soft (0.3%) or hard (1.5%) agar CYE plates (*Bustamante et al., 2004*).

The deletion strains and the strains expressing the different Neon Green fusions were obtained using a double-recombination strategy as previously described (*Bustamante et al., 2004*; *Shaner et al., 2013*). Briefly, the *kil* deletion alleles (carrying ~700-nucleotide long 5' and 3' flanking sequences of *M. xanthus* locus tags) were Gibson assembled into the suicide plasmid pBJ114 (*galK*, Kan^R) and used for allelic exchange. Plasmids were introduced in *M. xanthus* by electroporation. After selection, clones containing the deletion alleles were identified by PCR. Using the same strategy, 'Neon Green fusion' alleles were introduced at *kilD* and *kilF* loci. The corresponding strains expressed, under the control of their native promoters, N-terminal Neon Green fusions of KilD and KilF.

For complementation of Δ*kilC*, Δ*kilF*, Δ*kilG*, and Δ*kilH* strains, we used the pSWU19 plasmid (Kan^R) allowing ectopic expression of the corresponding genes from the *pilA* promoter at Mx8-att site. The same strains transformed with the empty vector were used as controls.

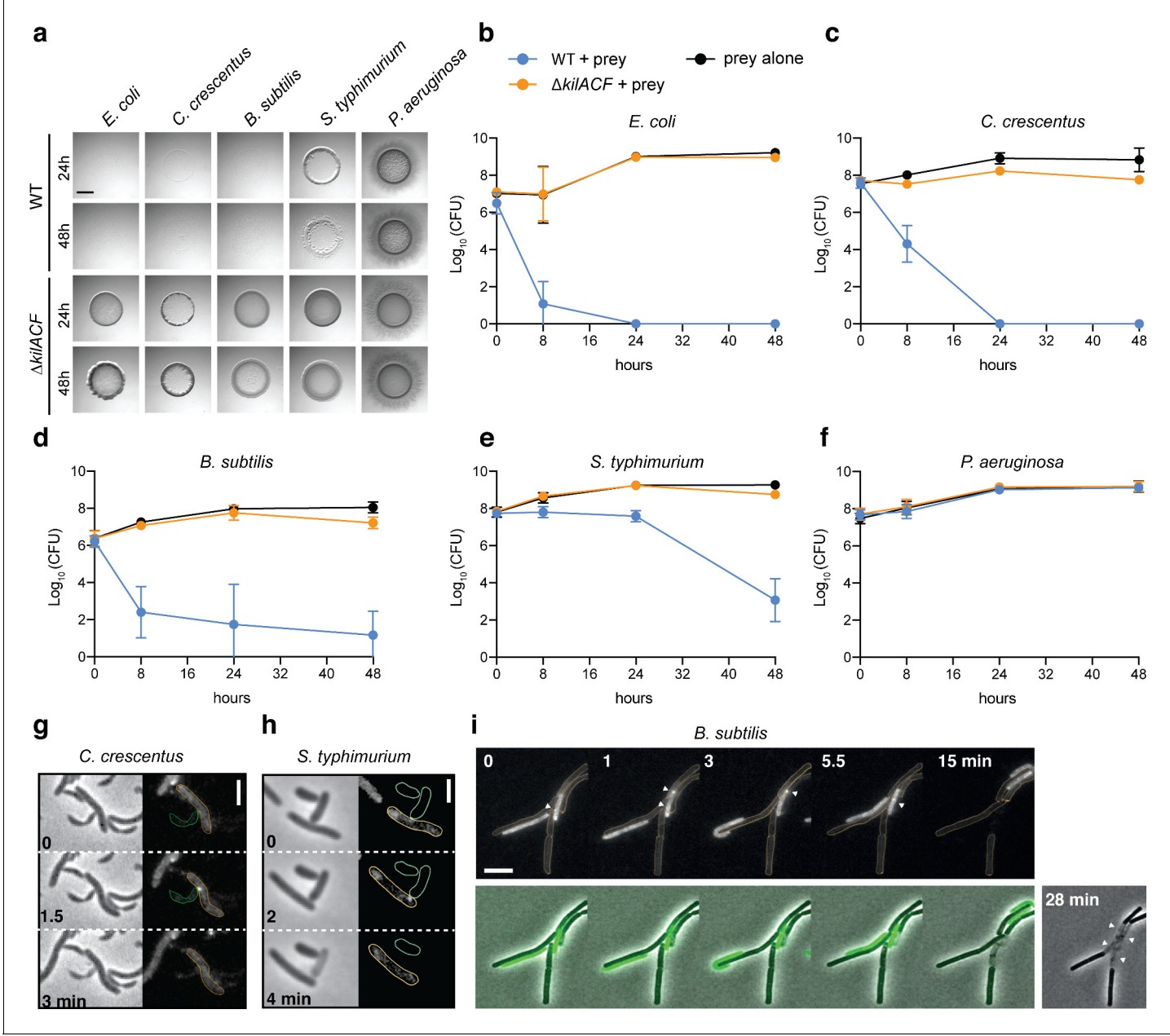

**Figure 6.** The Kil system mediates killing against diverse bacterial species. (a) The *kil* genes are predation determinants against various species. To evaluate if *M. xanthus kil* mutant had lost the ability to lyse by direct contact different preys, prey-cell suspensions were directly mixed with *M. xanthus* WT or Δ*kilACF* and spotted on CF agar (+ 0.07% glucose). After 24 and 48 hr of incubation, pictures of the spots corresponding to the different predator/prey couples were taken. Note that *Pseudomonas aeruginosa* is resistant in this assay. Scale bar = 3.5 mm. (b, c, d, e, f) Prey cell survival upon predation was evaluated by CFU counting. The different preys (Kan^R) were mixed with *M. xanthus* WT (blue circles) or Δ*kilACF* (orange circles) strains and spotted on CF agar (+ 0.07% glucose). Spots were harvested after 0, 8, 24, and 48 hr of predation, serially diluted and platted on agar plates with kanamycin for CFU counting. The prey alone (black circles) was used as a control. Two experimental replicates were used per time point. This experiment was independently performed three times. Error bars represent the standard deviation to the mean. (g) NG-KilD cluster formation and subsequent contact-dependent killing of *Caulobacter crescentus*. Scale bar = 2 μm. See corresponding *Video 11* for the full time lapse. (h) NG-KilD cluster formation and subsequent contact-dependent killing of *Salmonella enterica* Typhimurium. Scale bar = 2 μm. See corresponding *Video 12* for the full time lapse. (i) NG-KilD cluster formation and subsequent contact-dependent killing of *B. subtilis*. See corresponding *Video 13* for the full time lapse. Scale bar = 2 μm.

The online version of this article includes the following source data for figure 6:

**Source data 1.** Prey CFU counts during predation (*Figure 6b,c,d,e,f*).

To express KilG C-terminally fused to Neon Green, a pSWU19-*PpilA-kilG-NG* was created and transformed in the Δ*kilG* strain.

Western blotting was performed as previously described (*Bustamante et al., 2004*) using a commercial polyclonal anti Neon-Green antibody (Chromotek).

## Growth rate comparison in liquid cultures

To compare growth rates of *M. xanthus* WT and Δ*kilACF* strains, overnight CYE cultures were used to inoculate 25 ml of CYE at $OD_{600}$ = 0.05. Cultures were then incubated at 32°C with a shaking speed of 160 rpm. To avoid measuring cell densities at night, a second set of cultures were inoculated 12 hr later at $OD_{600}$ = 0.05. Every 4 hr, 1 ml sample of each culture was used to measure optical densities at 600 nm with a spectrophotometer. The different measurements were then combined into a single growth curve. This experiment was performed with three independent cultures per strain.

## Predation assay on agar plates

### Prey colony invasion on CF agar plates

*M. xanthus* and *E. coli* were respectively grown overnight in 20 ml of CYE at 32°C and in 20 ml of LB at 37°C. The next day, cells were pelleted and resuspended in CF medium (MOPS 10 mM pH 7.6; $KH_2PO_4$ 1 mM; $MgSO_4$ 8 mM; $(NH_4)_2SO_4$ 0.02%; Na citrate 0.2%; Bacto Casitone 0.015%) to a final $OD_{600}$ of 5. 10 µl of *M. xanthus* and prey cell suspensions were then spotted next to each other (leaving less than 1 mm gap between each spot) on CF 1.5% agar plates with or without 0.07% glu-

cose (to allow minimal growth of the prey cells) and incubated at 32°C. After 48 hr incubation, pictures of the plates were taken using a Nikon Olympus SZ61 binocular loupe (x10 magnification) equipped with a camera and an oblique filter. ImageJ software was used to measure the surface of the prey spot lysed by *M. xanthus*.

### Spotting predator-prey mixes on CF agar plates

To force the contact between *M. xanthus* and a prey, mixes of predator/prey were made and spotted of CF agar plates. 200 µl of a prey cell suspension (in CF, $OD_{600}$ = 5)were mixed with 25 µl of a *M. xanthus* cell suspension (in CF, $OD_{600}$ = 5) and 10 µl of this mix were spotted on CF agar plates supplemented with 0.07% glucose. As described above, pictures of the plates were taken after 24 hr incubation.

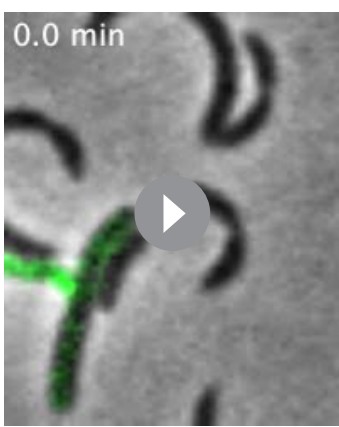

**Video 11.** NG-KilD cluster formation in contact with *Caulobacter crescentus*. Shown is an overlay of the fluorescence and phase contrast images of a motile *Myxococcus* cell in predatory contact with a *C. crescentus* cell. The movie was shot at ×100 magnification objective for 7 min. Pictures were taken every 30 s.
https://elifesciences.org/articles/72409#video11

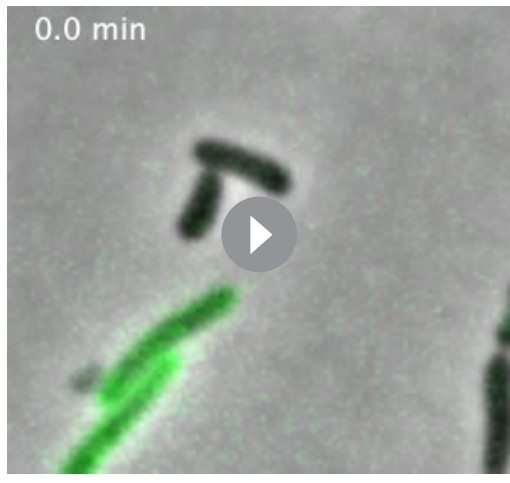

**Video 12.** NG-KilD cluster formation in contact with *Salmonella typhimurium*. Shown is an overlay of the fluorescence and phase contrast images of a motile *Myxococcus* cell in predatory contact with an *S. enterica* Typhimurium cell. The movie was shot at ×100 magnification objective for 20 min. Pictures were taken every 30 s.
https://elifesciences.org/articles/72409#video12

## Visualizing prey colony invasion and contact-dependent killing by microscopy

### Prey colony invasion on CF agar pads

Prey invasion was imaged by microscopy using the Bacto-Hubble system the specific details of the Method are described elsewhere (*Panigrahi, 2020*). Briefly, cell suspensions concentrated to $OD_{600}=5$ were spotted at 1 mm distance onto CF 1.5% agar pads and a Gene Frame (Thermo Fisher Scientific) was used to sandwich the pad between the slide and the coverslip and limit evaporation of the sample. Slides were incubated at 32°C for 6 hr before imaging, allowing *Myxococcus* and *E. coli* to form microcolonies. Time-lapse of the predation process was taken at ×40 or ×100 magnification. Movies were taken at the invasion front where *Myxococcus* cells enter the *E. coli* colony. To facilitate tracking, *M. xanthus* cells were labeled with fluorescence (*Ducret et al., 2013*). Fluorescence images were acquired by microscopy every 30 s for up to 10 hr, at room temperature (see below for experimental details of time lapse acquisitions ).

### Spotting predator-prey mixes on CF agar pads

To image contact-dependent killing between *M. xanthus* and prey cells (*E. coli, C. crescentus, B. subtilis, S. typhimurium, and P. aeruginosa*), cells were grown as described above, pelleted and resuspended in CF medium to a final $OD_{600}$ of 1. Equal volumes of *M. xanthus* and prey cell suspensions were then mixed together and 1 µl of the mix was spotted on a freshly made CF 1.5% agar pad on a microscope slide. After the spot has dried, the agar pad was covered with a glass coverslip, and incubated in the dark at room temperature for 20–30 min before imaging.

Time-lapse experiments were performed using two automated and inverted epifluorescence microscope: a TE2000- E-PFS (Nikon), with a ×100/1.4 DLL objective and an ORCA Flash 4.0LT camera (Hamamatsu) or a Ti Nikon microscope equipped with an ORCA Flash 4.0LT camera (Hamamatsu). Theses microscopes are equipped with the 'Perfect Focus System' (PFS) that automatically maintains focus so that the point of interest within a specimen is always kept in sharp focus at all times, despite any mechanical or thermal

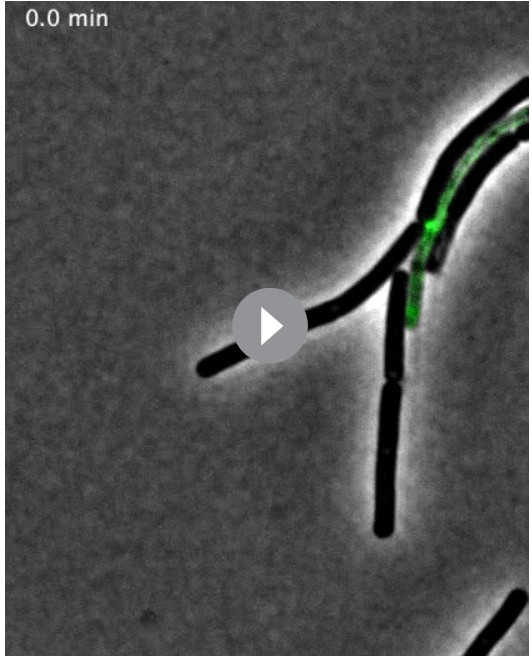

**Video 13.** NG-KilD cluster formation in contact with *Bacillus subtilis*. Shown is an overlay of the fluorescence and phase contrast images of a motile *Myxococcus* cell in predatory contact with a *B. subtilis* cell. The movie was shot at ×100 magnification objective for 30 min. Pictures were taken every 30 s.
https://elifesciences.org/articles/72409#video13

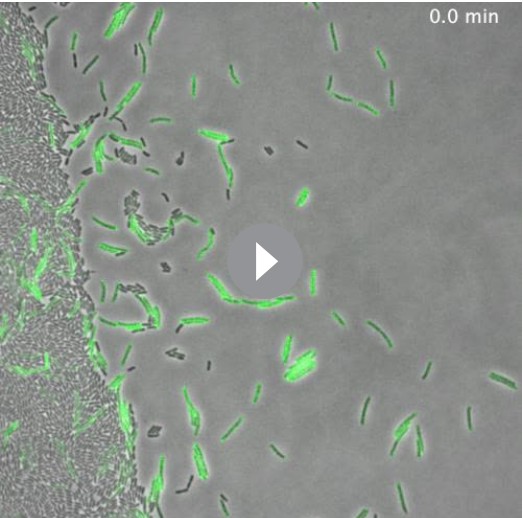

**Video 14.** *Pseudomonas aeruginosa* is not lysed by *Myxococcus* and does not induce NG-KilD cluster formation. Shown is an overlay of the fluorescence and phase contrast images of a motile *Myxococcus* cells mixed with *Pseudomonas* cells. The movie was shot at ×100 magnification objective for 30 min. Pictures were taken every 30 s.
https://elifesciences.org/articles/72409#video14

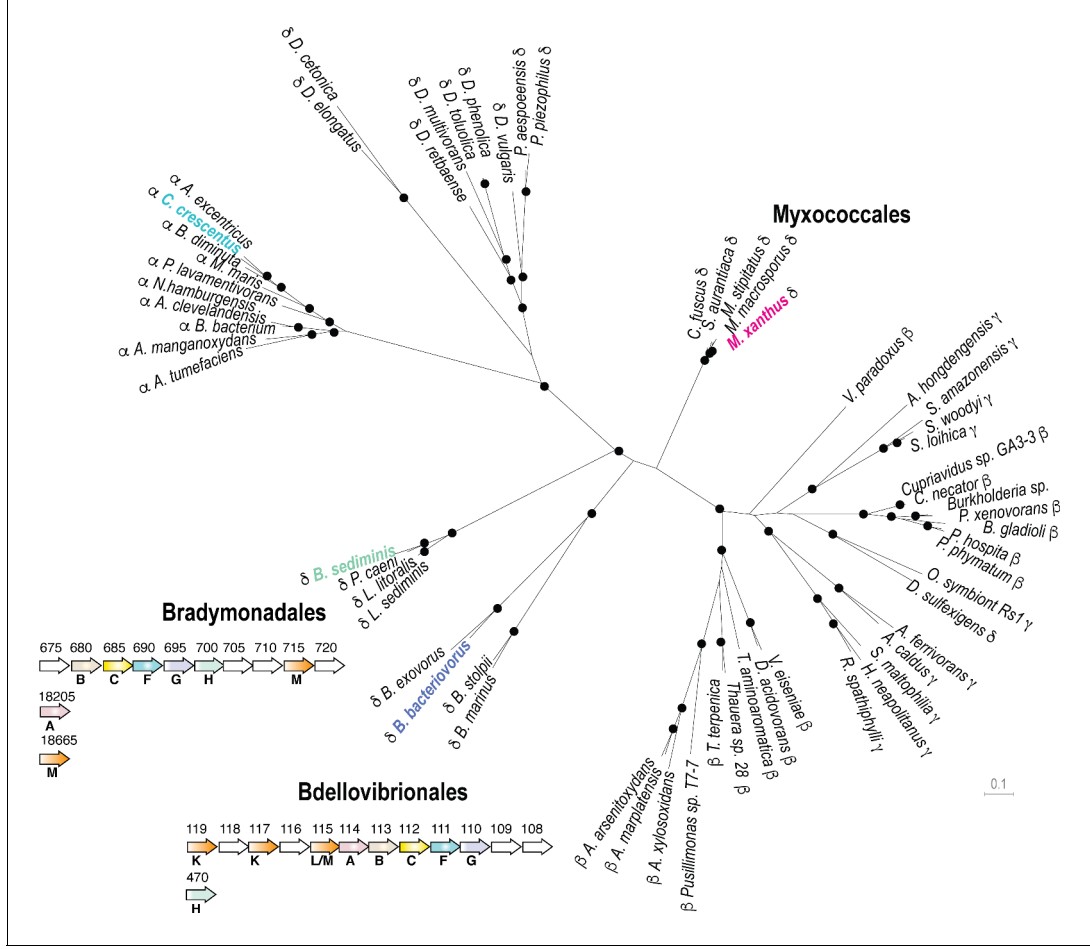

**Figure 7.** The Kil system is conserved in predatory delta-proteobacteria. Phylogenetic tree of the Type-IV filamentous system that gave rise to the *M. xanthus* Kil system. Only the four well-conserved Kil system components were used for constructing the phylogenetic tree. Dots indicate stable bootstrap values (> 75), classes are indicated next to species names. The *M. xanthus* Kil system is also found in other *Myxococcales* and closely related systems are also present in *Bradymonadales* and *Bdellovibrionales*, suggesting a functional specialization related to predation. The genetic organization of *kil*-like genes is shown for example members of each orders, *Bradymonas sediminis* and *Bdellovibrio bacteriovorus* (see also *Supplementary file 2*). The nomenclature and color code for Kil homologs are the same as in *Figure 3*. Gene accession numbers (KEGG) are shown above gene symbols.

The online version of this article includes the following source data and figure supplement(s) for figure 7:

**Source data 1.** Supermatrix alignment.
**Figure supplement 1.** Predation phenotype of a *Myxococcus* D,D-decarboxylase mutant (*Zhang et al., 2020a*).

perturbations. Images were recorded with NIS software from Nikon. All fluorescence images were acquired with appropriate filters with a minimal exposure time to minimize photo-bleaching and phototoxicity effects: 30 min long time-lapses (one image acquired every 30 s) of the predation process were taken at x100 magnification. DIA images were acquired using a 5 ms light exposure and GFP fluorescent images were acquired using a 100 ms fluorescence exposure with power intensity set to 50% (excitation wavelength 470 nm) to avoid phototoxicity.

## Labelling *E. coli* cells with the fluorescent D-amino acid TADA

Lyophilized TADA (MW = 381.2 g/mol, laboratory stock *Faure et al., 2016*) was re-suspended in DMSO at 150 mM and conserved at −20°C. The labeling was performed for 2 hr in the dark at room temperature, using 2 µl of the TADA solution for 1 ml of cells culture ($OD_{600}$ = 2). Cells were then washed four times with 1 ml of CF and directly used for predation assays on agar pad.

### Image analysis

Image analysis was performed under FIJI (*Schindelin et al., 2012*) and MicrobeJ (*Ducret et al., 2016*) an ImageJ plug-in for the analysis of bacterial cells.

## Semantic segmentation of *Myxococcus* cells

This was performed using the newly developed MiSiC system, a deep learning-based bacterial cell segmentation tool (*Panigrahi, 2020*). The system was used in semantic segmentation mode and annotated manually to reveal *E. coli* lysing cells.

## Kymograph construction

Kymographs were obtained after manual measurements of fluorescence intensities along FIJI hand-drawn segments and the FIJI-Plot profile tool. The measurements were then exported into the Prism software (Graphpad, Prism 8) to construct kymographs.

## Cell tracking

Cell tracking and associated morphometrics were obtained using MicrobeJ. Image stacks were first processed, stabilized and filtered with a moderate Gaussian blur and cells were detected by thresholding and fitted with the Plug-in 'medial axis' model. Trajectories were systematically verified and corrected by hand when necessary.

## Tracking *Myxococcus* motility pauses , NG-KilD foci formation and prey cell lysis during predation

In 30 min time-lapses, contacts between prey cells and *Myxococcus* cells were scored. Pauses were counted when the predatory cell stopped all movement upon contact with the prey. We also counted if these contacts lead to the formation of NG-KilD foci and to cell lysis. Thus, for a determined *E. coli* cell, we scored the number of contacts with *Myxococcus*, the number of pauses these contacts induces in *M. xanthus* motility, the number of NG-KilD foci formed upon contacts and, ultimately, the lysis of the cell. Five independent movies were analyzed for each strain and the percentage of contacts leading to a pause in motility, NG-KilD foci formation and cell lysis was calculated. We also estimated the percentage of NG-KilD clusters leading to cell lysis.

## Tracking cluster time to lysis

Time to lysis measures the elapsed time between cluster appearance to prey cell death. Data were obtained from two biological replicates.

### CPRG assay for contact-dependent killing in liquid

## CPRG assay in 24-well plates

*M. xanthus* and *E. coli* cultures were grown overnight, pelleted and resuspended in CF at $OD_{600}$ ~5. 100 µl of *M. xanthus* cell suspension (WT and mutants) were mixed with 100 µl of *E. coli* cell suspension in a 24-well plate. In each well, 2 ml of CF medium supplemented with CPRG (Sigma Aldrich, 20 µg/ml) and IPTG (Euromedex, 50 µM) were added to induce *lacZ* expression. The plates were then incubated at 32°C with shaking and pictures were taken after 24 and 48 hr of incubation. To test the contact-dependance, a two-chamber assay was carried out in a Corning 24 well-plates containing a 0.4 µm pore polycarbonate membrane insert (Corning Transwell 3413). This membrane is permeable to small metabolites and proteins and impermeable to cells. *E. coli* cells were inoculated into the top chamber and *M. xanthus* cells into the bottom chamber.

## CPRG assay in 96-well plates

To evaluate the predation efficiency of the different *kil* mutants, the CPRG assay was adapted as follow: wild-type *M. xanthus* and the *kil* mutant strains were grown overnight in 15 ml of CYE. *E. coli* was grown overnight in 15 ml of LB. The next morning, *M. xanthus* and *E. coli* cells were pelleted and resuspended in CF at $OD_{600}$ = 0.5 and 10, respectively. To induce expression of the β-galactosidase, IPTG (100 µM final) was added to the *E. coli* cell suspension.

In a 96-well plate, 100 µl of *M. xanthus* cell suspension were mixed with 100 µl of *E. coli* cell suspension. Wells containing only *M. xanthus*, *E. coli* or CF were used as controls. The lid of the 96-well plate was then sealed with a breathable tape (Greiner bio-one) and the plate was incubated for 24 hr at 32°C while shaking at 160 rpm. In this setup, we observed that *M. xanthus* and *E.coli* cells aggregate at the bottom of the well and therefore come in direct contact, favoring predation in liquid.

The next day, the plate was centrifuged 10 min at 4800 rpm and 25 µl of the supernatant were transferred in a new 96-well plate containing 125 µl of Z-buffer (Na$_2$HPO$_4$ 60 mM, NaH$_2$PO$_4$ 40 mM, KCl 10 mM pH7) supplemented with 20 µg/ml of CPRG. After 15–30 min of incubation at 37°C, the enzymatic reaction was stopped with 65 µl of Na$_2$CO$_3$ (1 M) and the absorbance at 576 nm was measured using a TECAN Spark plate reader.

This experiment was performed independently four times. For Miller unit calculation, after absorbance of the blank (with CF) reaction was subtracted, the absorbances measured at 576 nm were divided by the incubation time and the volume of cell lysate used for reaction. The resulting number was then multiplied by 1000.

## Crystal violet biofilm staining

In a 96-well plate, 100 µl of *M. xanthus* cell suspension (in CF, OD$_{600}$ = 0.5) were mixed with 100 µl of *E. coli* cell suspension (in CF, OD$_{600}$ = 10) and incubated for 24 hr at 32°C while shaking at 160 rpm. The next day, the supernatant was carefully removed and the wells were washed with 200 µl of CF twice. Then, 100 µl of a 0.01% crystal violet solution were added to each well and incubated for 5 min. Wells were washed twice with 200 µl of water before imaging.

## Prey CFU counting after predation

*E. coli, S. typhi, P. aeruginosa,* and *B. subtilis* kanamycin resistant strains were grown at 37°C in liquid LB supplemented with kanamycin (50 or 10 µg/ml). *C. crescentus* kanamycin resistant strain was grown at 30°C in liquid PYE supplemented with kanamycin (25 µg/ml). Wild-type and Δ*kilACF M. xanthus* strains were grown at 32°C in liquid CYE. Cells were then centrifuged and pellets were resuspended in CF at an OD$_{600}$ of 5. 25 µl of *M. xanthus* cell suspensions and 200 µl of prey cell suspensions were then mixed together and 10 µl were spotted on CF agar plates supplemented with 0.07% glucose. After drying, plates were incubated at 32°C. At 0, 8, 24, and 48 hr time points, spots were harvested with a loop and resuspended in 500 µl of CF. This solution was then used to make 10-fold serial dilutions in a 96-well plate containing CF. At the exception of *C. crescentus*, 5 µl of each dilution were spotted on LB agar plates supplemented with 10 µg/ml of kanamycin and incubated at 37°C for 24 hr. *C. crescentus* dilutions were spotted on PYE agar plates supplemented with 25 µg/ml of kanamycin and incubated at 30°C for 24 hr. The next day, colony-forming units were counted and the number of prey cells that survived in the predator/prey spot was calculated.

## Measurements of *E. coli* killing by flow cytometry

*M. xanthus* strains (wild-type and *kil* mutants) constitutively expressing GFP were grown overnight in liquid CYE without antibiotics. *E. coli* mCherry (prey) was grown overnight in liquid LB supplemented with ampicillin (100 µg/ml). The next morning, optical densities of the cultures were adjusted in CF medium to OD$_{600}$=5. *M. xanthus* GFP and *E. coli* mCherry cell suspensions were then spotted onto fresh CF 1.5% agar plates as previously described (*Bustamante et al., 2004*). Briefly, 10 µl drops of the prey and the predator cell suspensions were placed next to each other and let dry. Inoculated plates were then incubated at 32°C. Time 0 corresponds to the time at which the prey and the predator spots were set on the CF agar plate. At time 0, 24, 48, and 72 hr (post predation) and for each *M. xanthus* strain, two predator/prey spot couples were harvested with a loop and resuspended in 750 µl of TPM. To fix the samples, paraformaldehyde (32% in distilled water, Electron Microscopy Sciences) was then added to the samples to a final concentration of 4%. After 10 min incubation at room temperature, samples were centrifuged (8 min, 7500 rpm), cell pellets were then resuspended in fresh TPM and optical densities were adjusted to OD$_{600}$ ~0.1.

Samples were then analyzed by flow cytometry on a Bio-Rad S3e Cell Sorter and data were processed using ProSort and FlowJo softwares. For each sample, a total population of 500,000 events was used and corresponds to the sum of *M. xanthus*-GFP and *E. coli*-mCherry events. A blue laser

(488 nm, 100 mW) was used for detection of forward scatter (FSC) and side scatter (SSC) and for excitation of GFP. A yellow-green laser (561 nm, 100 mW) was used for excitation of mCherry. GFP and mCherry signals were collected using, respectively, the emission filters FL1 (525/30 nm) and FL3 (615/25 nm) and a compensation was applied on the mCherry signal. Samples were run using the low-pressure mode (~10,000 particles/s). The density plots obtained (small angle scattering FSC versus wide angle scattering SSC signal) were first gated on the overlapped populations of *M. xanthus* and *E. coli*, then filtered to remove the multiple events and finally gated for high FL1 signal (*M. xanthus*-GFP) and high FL3 signal (*E. coli*-mCherry).

## Bioinformatics analyses

### Homology search strategy

We used several search strategies to identify all potential homologous proteins of the Kil system: we first used BLAST (*Camacho et al., 2009*; *Altschul et al., 1997*) to search for reciprocal best hits (RBH) between the *M. xanthus* and the *B. bacteriovorus* and *B. Sediminis* Kil systems, as well as the *C. crescentus* Tad system, identifying *bona fide* orthologs between the three species. We limited the search space to the respective proteomes of the three species. We then used HHPRED (*Hildebrand et al., 2009*) to search for remotely conserved homologs in *B. bacteriovorus* using the proteins from the two operons identified in *M. xanthus*. Finally, we performed domain comparisons between proteins from the *B. bacteriovorus* and *B. sediminis* Kil operons and *C. crescentus* Tad system to identify proteins with similar domain compositions in *M. xanthus*. Identified orthologs or homologs between the three species, the employed search strategy, as well as resulting e-values are shown in *Supplementary file 2* (Table 2): *M. xanthus* proteins with homologs identified in *B. bacteriovorus* HD100, *C. crescentus* CB15 and *B. sediminis*.

### Structure predictions

Tertiary structural models of the secretin and the cytoplasmic ATPase were done using Phyre2 (*Kelley et al., 2015*) or SWISS-MODEL (*Waterhouse et al., 2018*), in both cases using default parameters. Quaternary models were generated using SWISS-MODEL. Structural models were displayed using Chimera (*Pettersen et al., 2004*) and further processed in Illustrator .

### Phylogenetic analyses

We used the four well-conserved Kil system components for phylogenetic analysis. To collect species with secretion systems similar to the Kil system, we first used MultiGeneBLAST (*Medema et al., 2013*) with default parameters. Orthologs of the four proteins from *B. bacteriovorus*, *B. Sediminis* and *C. crescentus* from closely related species were added manually. We aligned each of the four proteins separately using MAFFT (*Katoh et al., 2002*) and created a supermatrix from the four individual alignments. Gblocks (*Katoh et al., 2002*) using relaxed parameters was used prior to tree reconstruction to remove badly aligned or extended gap regions. . Alignments of individual trees were also trimmed using Gblocks. PhyML (*Guindon et al., 2010*) was used for tree reconstruction, using the JTT model and 100 bootstrap iterations. Trees were displayed with Dendroscope (*Huson and Scornavacca, 2012*) and further processed in Illustrator .

## Acknowledgements

We thank Lotte Søgaard-Andersen and Anke Treuner-Lange for the gift of the VipA plasmid. We thank Laurent Aussel for *E. coli* plasmids, Anne Galinier lab for the *Bacillus subtilis* strains, Emanuele Biondi lab for the *Caulobacter crescentus* strains and Sophie Bleves for the *Pseudomonas aeruginosa* strain. We thank Dorothée Murat, Romé Voulhoux, Marcelo Nöllmann, Vladimir Pelicic and Friedhelm Pfeiffer for discussions. Research in TM lab was supported by a 2019 CNRS 80-Prime allowance on bacterial predation and pattern formation. SS, PDB and DR are supported by an MENRT thesis grant from the ministry of research.

## Additional information

### Competing interests

Tâm Mignot: Reviewing editor, *eLife*. The other authors declare that no competing interests exist.

### Funding

| Funder | Grant reference number | Author |
|---|---|---|
| Centre National de la Recherche Scientifique | 2019 CNRS 80-Prime | Tâm Mignot |
| Ministère de l'Education Nationale, de la Formation professionnelle, de l'Enseignement Supérieur et de la Recherche Scientifique | MENRT thesis grant | Sofiene Seef Paul de Boissier Donovan Robert |

The funders had no role in study design, data collection and interpretation, or the decision to submit the work for publication.

### Author contributions

Sofiene Seef, Rikesh Jain, Conceptualization, Formal analysis, Validation, Investigation, Visualization, Methodology, Writing - review and editing; Julien Herrou, Bianca H Habermann, Conceptualization, Formal analysis, Supervision, Validation, Investigation, Visualization, Methodology, Writing - review and editing; Paul de Boissier, Donovan Robert, Formal analysis, Validation, Investigation, Visualization, Methodology, Writing - review and editing; Laetitia My, Investigation, Methodology, Writing - review and editing; Gael Brasseur, Conceptualization, Formal analysis, Validation, Methodology, Writing - review and editing; Romain Mercier, Formal analysis, Investigation, Writing - review and editing; Eric Cascales, Formal analysis, Investigation, Methodology, Writing - review and editing; Tâm Mignot, Conceptualization, Resources, Formal analysis, Supervision, Funding acquisition, Validation, Investigation, Visualization, Methodology, Writing - original draft, Project administration, Writing - review and editing

### Author ORCIDs

Julien Herrou (ID) https://orcid.org/0000-0002-8585-8043
Laetitia My (ID) http://orcid.org/0000-0002-7876-1809
Eric Cascales (ID) http://orcid.org/0000-0003-0611-9179
Bianca H Habermann (ID) http://orcid.org/0000-0002-2457-7504
Tâm Mignot (ID) https://orcid.org/0000-0003-4338-9063

### Decision letter and Author response

Decision letter https://doi.org/10.7554/eLife.72409.sa1
Author response https://doi.org/10.7554/eLife.72409.sa2

## Additional files

### Supplementary files

• Supplementary file 1. Table 1: Characteristics of *Myxococcus xanthus* proteins.

• Supplementary file 2. Table 2: *M. xanthus* proteins with homologs identified in *B. bacteriovorus* HD100, *C. crescentus* CB15 and *B. sediminis*.

• Supplementary file 3. Table 3: Strains.

• Supplementary file 4. Table 4: Plasmids.

• Supplementary file 5. Table 5: Primers.

• Transparent reporting form

## Data availability

Source data files have been provided for Figure 2—source data 1 -*E. coli* loss of fluorescence during contact-dependent lysis (Figure 2c); Figure 2—figure supplement 5—source data 1 - Contact dependent-lysis and VipA-GFP dynamics; Figure 2—figure supplement 7—source data 1 - CPRG assay; Figure 3—source data 1 -CPRG assay (Figure 3b); Figure 3—source data 2 - Counting percentage of contacts with a prey leading to motility pauses and prey cell lysis (Figure 3c, 3d); Figure 3—figure supplement 3—source data 1 - CPRG assay; Figure 4—source data 1 - Counting percentage of contacts with a prey leading to NG-KilD foci formation and counting percentage of NG-KilD foci associated with motility pause and prey cell lysis (Figure 4e, 4f, 4g); Figure 4—figure supplement 1—source data 1 -CPRG assay; Figure 4—figure supplement 2—source data 1 -CPRG assay; Figure 4—figure supplement 3—source data 1 -Lysis time; Figure 4—figure supplement 4—source data 1 - Western Blot; Figure 5—source data 1 - Flow cytometry (Figure 5c, 5d); Figure 5—source data 2 -M. xanthus growth during prey colony invasion (Figure 5e); Figure 5—source data 3 -Increase in M. xanthus cell length during predation (Figure 5f); Figure 5—figure supplement 2—source data 1 -Growth curves; Figure 6—source data 1 -Prey CFU counts during predation (Figure 6b,c,d,e,f); Figure 7—source data 1 -Supermatrix alignment; Figure 3—figure supplement 2 -RNA-seq Data from Livingstone PG et al. (2018) Microb Genom. PMID:29345219, Supplementary File 1 available online: https://www.microbiologyresearch.org/content/journal/mgen/10.1099/mgen.0.000152#supplementary_data.

The following previously published dataset was used:

| Author(s) | Year | Dataset title | Dataset URL | Database and Identifier |
|---|---|---|---|---|
| Livingstone PG, Millard AD, Swain MT, Whitworth DE | 2018 | unmapped read data | https://www.ncbi.nlm.nih.gov/sra/?term=PRJNA408275 | NCBI Sequence Read Archive, PRJNA408275 |

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
