## [Decision Letter]

**Acceptance summary:**

Myxococcus xanthus is well known for its social behaviors and predatory activity toward other bacteria, but precisely how it kills other bacteria has not been well characterized. It has often been assumed that this involves secreted, diffusible factors like antibiotics. This paper presents a fascinating set of observations that reveal a pilus/secretion system in Myxococcus that is necessary for a contact-dependent predation of other bacteria. The work presented lays the foundation for future studies of this contact-dependent predation.

**Decision letter after peer review:**

[Editors’ note: the authors submitted for reconsideration following the decision after peer review. What follows is the decision letter after the first round of review.]

Thank you for submitting your work entitled "A Tad-like apparatus is required for contact-dependent prey killing in predatory social bacteria" for consideration by *eLife*. Your article has been reviewed by 3 peer reviewers, and the evaluation has been overseen by a Reviewing Editor and a Senior Editor. The following individuals involved in review of your submission have agreed to reveal their identity: Lotte Sogaard-Andersen (Reviewer #1); Alain Filloux (Reviewer #2); John Kirby (Reviewer #3).

We are sorry to say that, after consultation with the reviewers, we have decided that your work, as it stands now, cannot be considered further for publication by *eLife*. The reviewers all appreciated the potential novelty of the findings, but also highlighted several key shortcomings. Some of these issues were raised in the initial reviews conducted by Nature Microbiology and transferred to *eLife*. Importantly, some of the reviewers this time were in common with those from the first round and they did not feel that the issues raised had been adequately addressed. In particular, the reviewers agreed that the paper does not convincingly provide (i) a clear demonstration that a Tad-like pilus is produced by Myxococcus, and (ii) evidence that the pilus assembles concomitantly and spatially coincident with the KilD clusters. Data addressing these concerns were deemed critical as the paper also does not address whether the pilus is driving toxin delivery or is simply involved in promoting proximity to prey. The reviewers unanimously agreed that a highly revised paper that fully addressed these issues could warrant reconsideration. However, because these important points had not been addressed in this revision after they were raised in the first round of reviews, there was a sense that you may not want to or may not be experimentally able to address them, hence the final decision to reject.

*Reviewer #1:*

In this manuscript, the authors explore mechanisms involved in the predation of other bacteria by Myxococcus xanthus. The major findings are (1) M. xanthus cells depend on gliding motility to efficiently invade an *E. coli* prey colony. (2) *E. coli* prey cells are lysed in a contact-dependent manner. (3) When M. xanthus cells make prey contact, they sometimes pause and then kill the prey cell. (4) Using a genetic screen, two gene clusters (referred to as the kil gene clusters) are identified that encode proteins, some of which have homology to those of Tad pili. Some of the Kil proteins are important for pausing of cells and killing of prey. (5) One of the suggested Kil proteins assemble to form clusters upon prey contact; however, assembly of these clusters is independent of other Kil proteins On the basis of these findings the authors suggest that the Kil proteins assemble to form a Tad pilus system and are important for pausing and prey killing. Overall, this is an interesting manuscript; however, it remains unclear what the actual function of the identified Kil proteins are.

Weaknesses include

(1) The lack of genetic complementation experiments. Thus, it is unclear precisely which of the Kil proteins are important for predation.

(2) The Kil proteins are encoded in two gene clusters. The evidence that these proteins make up a Tad pilus system is based on homology and that mutations in both clusters result in reduced predation. No evidence is presented that proteins encoded by these two clusters interact to form a Tad pilus machine.

(3) The authors localize the Kil system using an NG-KilD fusion; however, there is no evidence that KilD, which is a FHA domain-containing protein, associates with the Tad pilus machinery. In fact, KilD makes clusters independently of all other Kil proteins tested suggesting that these clusters may not report on Kil assembly and activity. An equally plausible scenario is that Myxococcus/*E. coli* contacts result in activation of KilD leading to the formation of foci. These foci then signal assembly of the Kil system somewhere in a cell (or maybe not). Therefore, it is not clear where and if this machinery localizes during prey contact.

(4) I did not find a description of how the mutagenesis was done. Please include a description of how the mutagenesis was done, how many mutants were screened, and in which loci the mutations (transposon insertions?) occurred. Was the screen saturated?

(5) Throughout the manuscript, the authors need to tone down their conclusions and stick to what they actually show. It is also important that the authors present their results in the context of what is already known about contact-dependent killing in M. xanthus.

(6) Line 20: delete "perhaps". It is well established from the work in ref. 7 that killing can occur in a contact-dependent manner.

(7) Line 22-23: There is no evidence in the manuscript that killing is driven by the Kil system. The evidence presented shows that the Kil system is important for killing.

(8) Line 24: Strictly speaking the authors have shown that one Kil protein assembles at the contact site.

(9) Line 66 and 69: It has been known for more than 20 years that type IV pili and exopolysaccharide are important for S-motility. Please include the relevant references.

(10) Line 90-91 and Figure 2A: It is difficult/impossible to see the lysed cells in the upper panel in Figure 2A. Would it be possible to label the individual prey cells with a number or in different colors to be able to follow cells over time?

(11) Line 91: Delete "remarkably" because this has been seen in ref. 7.

(12) Line 103-105: This detailed description fits to the "lower" *E. coli* cell but not to the "upper" *E. coli* cell in Figure 2D. Is that important? How many cells were analyzed? Is the cell in Figure 2E representative?

(13) Line 124: Because the killing mechanism is not identified, I suggest to rewrite to emphasize that you looked for genes important for killing.

(14) Line 124-131: As described, my understanding is that M. xanthus and *E. coli* cells are mixed in suspension culture and with shaking. From the data in Figure 1 and 2, one gets the impression that predator and prey have to be in close contact for several minutes before the *E. coli* prey lyses. How would this happen in suspension culture with shaking?

(15) Figure 4A, B: What precisely is shown in these two figures? And how are the figures different?

(16) Line 204-205: How do the experiments described lead to the conclusion that the "…unambiguously that cluster formation reflects contact-dependent killing"? Is it not more correct to say that your data support that cluster formation correlates with killing?

(17) Figure 4C and accompanying text: In how many cells was the reported pattern observed?

(18) Line 213: There is not a 2-reduction is foci formation according to figure 4D.

(19) Figure 5D: What do the red and green lines indicate?

*Reviewer #2:*

This works has clear novelty and describes aspects on how Myxococcus xanthus can kill other bacterial cells.

It starts by demonstrating that A-motility (Agl-Glt system) is needed to invade adjacent colony. These A-motile cells are able to kill a prey (*E. coli*) likely by making holes in the peptidoglycan layer, but the killing does not directly involve the A-motility system.

Based on this, the authors did use an elegant approach to identify mutants that can invade but cannot kill. Reported hits lie within two gene clusters encoding Tad proteins, which in other bacteria such as *Pseudomonas aeruginosa* are involved in the assembly of a Tad pilus which promotes bacterial attachment. The cluster 1 encodes the prepilin peptidase, the secretin and the ATPase, while cluster 2 encodes the inner membrane platform, major and minor pilins. Both clusters encode additional genes of unknown function. It is then shown that this Tad-like system, called Kil system, can trigger target cell lysis probably via the recruitment of other systems allowing delivery of toxic elements into prey cells. Despite not having data supporting what could contribute to the toxicity, it is shown that the Kil system is actually assembling at the site of contact with the prey cell. Finally, the authors also showed that the Myxococcus Kil system allows killing of a wide range of bacteria which are not necessarily phylogenetically-related.

In conclusion, this work brought novel and original concepts, some of which would definitively deserve further investigation in subsequent studies.

This is a paper that I previously reviewed for another journal.

Most of the queries I had have been addressed experimentally or discussed in the text. This includes further repeats on a number of experiments which are now providing statistically significant data supporting solid conclusions, including revisiting the killing of Bacillus.

There are still pending questions such as what is the effective mechanism which leads to killing upon Tad-dependent contact, is there a Tad-like pilus assembled or why is the KilA prepilin peptidase not essential for killing. I would agree that these are likely beyond the scope of the present study. Now writing "mysterious mechanism" in the abstract (line 20) might give the feeling that the story is incomplete, which should be avoided.

Another yet intriguing question would be whether some of the genes of unknown function encoded within each of the tad-like clusters are involved in killing.

*Reviewer #3:*

The authors set out to identify factors involved in bacterial predation and have identified a set of genes that are homologous to secretion systems. The study focuses on M. xanthus as the predator with *E. coli* as the prey source. The authors demonstrate that M. xanthus A-Motility (focal adhesion mediated motility) is required for efficient penetration of *E. coli* under the conditions of their assays. The authors identified two genes that encode proteins thought to assemble into a type IV filament-like machine, designated "Kil". This system inhibits motility of prey cells and stimulates lysis of those cells. They show that the Kil apparatus assembles near contact sites between predator and prey cells using protein-fusion constructs. However, there are limitations based on experimental design that diminish support for the stated conclusions. The assay used to identify genes of interest was conducted in aqueous media where the motility system of interest is not required. Furthermore, the microscopic techniques used here do not allow for the visualization of pores or precise localization of machinery thought to be involved in the mechanism for delivery of toxins from predator to prey cells.

I enjoyed reading this story but have several points that I would like to see addressed in a modified manuscript.

Results

Line 72-82.

The authors explore prey invasion by M. xanthus wild type, pilA and aglQ mutants. The aglQ strain did not efficiently penetrate the prey while the pilA did. However, Figure 1 shows the wild type and the aglQ strain. I would like to see the pilA strain for comparison. Similarly, the Figure legends for Video 1 and 2 are the same; it is not clear which cells were labeled with mCherry.

Line 91/92.

Similar results have been reported widely in the field, most recently by Zhang et al., 2020, and those efforts should be acknowledged. The statement that A-motile cells carry a prey toxic activity suggests that cells utilizing different motility systems might utilize different toxins for predation. Is that correct?

In Video 3 many prey cells are in contact with the predator, yet just only a small set of prey cells seemed to be lysed. This should be discussed by the authors.

Figure 2d and Line 538: "Holes in the PG-labelling are observed at the contact site" seems to be too strong. The figure I have here does not provide enough high resolution to conclude that "holes" are visible.

Line 124-134.

The authors use a liquid predation assay to monitor lysis of *E. coli*. Upon lysis β-galactosidase is released into the media that contains CPRG substrate that is converted into a dark red substance. This is an effective assay. However, A-motility as reported to date requires a solid surface that is not present in this type of assay. Thus, the results as reported are done in an A-motility independent manner, while is suggested otherwise throughout the manuscript. Also the data for all strains tested in parallel should be shown (WT vs aglQ vs pilA). Importantly, by definition, predation includes consumption and growth of prey. Thus, colony forming units should be reported in the presence and absence of each predator strain and for the prey.

Line 134/135.

The authors use the liquid assay to screen for mutations in the predicted cell-envelope complexes, in which contact-dependent killing is abolished. However, I could not find an adequate description in the Results or the Materials and methods on how this screen was conducted. Additionally, the two location of the two mutations identified should be stated.

Figure 3A shows the predicted model of the Tad-like apparatus. It is not clear that this apparatus is actually made.

Figure 6a, Line 620-622.

The authors state that the ∆kilACF cells move over the top of the *B. subtilis* and S. typhimurium colonies without lysis. It appears to me that lysis is occuring. Again, predation (killing, consumption and growth) should be quantified by colony forming units.

Line 317/318.

Epistasis analysis is typically required to establish genetic elements within a pathway. Additional biochemical evidence could be used to support the order of events and protein interactions. That evidence is not provided here.

Is M. xanthus resistant to its own Kil system?

[Editors’ note: further revisions were suggested prior to acceptance, as described below.]

Thank you for resubmitting your work entitled "A Tad-like apparatus is required for contact-dependent prey killing in predatory social bacteria" for further consideration by *eLife*. Your revised article has been evaluated by Gisela Storz (Senior Editor) and a Reviewing Editor.

The manuscript has been improved but there are some remaining issues that need to be addressed, as outlined below:

This revised version of the manuscript was seen by one of the original reviewers who is now satisfied with the latest round of changes made. The other two reviewers were not available for re-review, so a new reviewer has been enlisted. As you'll see, this reviewer is generally enthusiastic about the work, but raises some questions and makes some suggestions that we think are reasonable and will improve the manuscript further. Once made, we anticipate the paper then being accepted for publication.

Essential revisions:

*Reviewer #1:*

This is a revised version of a manuscript that I have previously seen.

The authors have included several additional experiments and adequately addressed all my comments. I look forward to seeing this manuscript published.

*Reviewer #2:*

In this study, the authors provide further evidence that predation by Myxococcus xanthus likely occurs primarily via a contact-dependent mechanism and identify one key player. They show that the motility/adhesion requirements for killing are dependent on the environment: type IV pili-mediated twitching group motility in liquid medium vs. solitary adventurous (A-type) propulsion on solid surfaces. By testing mutants in a short list of candidates (putative envelope-associated complexes), the authors identify two loci necessary for contact-dependent predation in both liquid and solid media. Primarily based on time-lapse microscopy of selected mutants, the manuscript presents sufficient evidence to support the central hypothesis that both loci are likely part of the same Tad-like pilus system (kil), and that this kil locus is involved in attachment and killing of prey cells. The exact role of kil in the process remains to be determined and it is likely that other yet unidentified factors are also necessary.

The authors are for the most part cautious to not draw conclusions beyond what is supported by the data and acknowledge the remaining gaps in knowledge. However, most of the conclusions of the study hinge on the analysis of fluorescent protein foci in time-lapse videos. While the authors provide quantification of several fields, most of the figures and videos only show one or maximum two cells displaying the behavior under study. Moreover, there is no "negative control" for the videos of KilD/F/G and AglZ (A-motility) contacts, to rule out that it's not just random transient aggregation by these labelled proteins (or that contact with other cells causes temporary physical changes in those membrane regions creating some sort of artefactual foci).

Another weakness is that I don't think the Tad phylogeny in Figure 7 supports the strong claim that "the kil genes evolved in predatory bacteria" (l. 298) or that the Tad homologs in predatory bacteria (Myxococcales, Bdellovibrionales and Bradymonadales) are a uniquely distinct clade of kil genes relative to the more distant tad homologs in other bacteria (my interpretation of l. 303-306). The Myxococcales, Bdellovibrionales and Bradymonadales tad clades are not monophyletic (as a group) as the authors seem to imply, so "suggesting a similar function" (l. 306) that's unique to these classes seems too speculative. Also, since the predatory bacteria are all δ-proteobacteria (and hence more closely related amongst them than the other clades on the phylogeny), wouldn't it be expected that their pili genes would be more similar, regardless or function? I therefore think these conclusions should be toned down. It would also be interesting for the authors to discuss if there are any predatory bacteria (or non-predatory δ-proteobacteria) where kil/tad homologues are absent.

– Timestamps in the videos would be extremely helpful.

– For a reader unfamiliar with the field, the introduction does not provide enough context to understand the nuances about what predation is and how it distinguishes from other types of antagonism. One short explanatory sentence at the beginning would be very helpful. In particular, the findings from reference 7 should be explained a little bit more.

– At the beginning of the discussion (l. 322-325), the authors should acknowledge contact-dependent plasmolysis has been previously described (ref. 7).

– I think it is very important to show videos of many fields/cells instead of just one representative cell, to show that the videos were not cherry-picked. Moreover, for the videos of KilD/F/G and AglZ (A-motility) contacts, I think it is important to show a negative control. Since the authors mention that a deletion in other kil components does not preclude foci formation (and therefore the mutant is not a control), maybe a good control are the *P. aeruginosa* videos?

– I understand that it may be technically complex to do time-lapse imaging of differently labeled kil components (e.g. KilF-GPF and KilG-mCherry). However, I think showing co-localization of the foci for the two components in a single time point (which would allow longer exposure times and large numbers of cells analyzed simultaneously) would provide more compelling evidence that both proteins are part of the same complex. Similarly, it would be interesting to see if KilD/F foci colocalize with AglZ.

– Do any of the other genes in the two loci have sequence signatures of toxins? (For instamce, to the HMMs compiled in Zhang D, de Souza RF, Anantharaman V, Iyer LM, Aravind L. Polymorphic toxin systems: Comprehensive characterization of trafficking modes, processing, mechanisms of action, immunity and ecology using comparative genomics. Biol Direct. 2012;7:18.)

– L. 232 there is a typo; should read "degradation was detected".

---

## [Author Response]

[Editors’ note: the authors resubmitted a revised version of the paper for consideration. What follows is the authors’ response to the first round of review.]

Reviewer #1:In this manuscript, the authors explore mechanisms involved in the predation of other bacteria by Myxococcus xanthus. The major findings are (1) M. xanthus cells depend on gliding motility to efficiently invade an *E. coli* prey colony. (2) *E. coli* prey cells are lysed in a contact-dependent manner. (3) When M. xanthus cells make prey contact, they sometimes pause and then kill the prey cell. (4) Using a genetic screen, two gene clusters (referred to as the kil gene clusters) are identified that encode proteins, some of which have homology to those of Tad pili. Some of the Kil proteins are important for pausing of cells and killing of prey. (5) One of the suggested Kil proteins assemble to form clusters upon prey contact; however, assembly of these clusters is independent of other Kil proteins On the basis of these findings the authors suggest that the Kil proteins assemble to form a Tad pilus system and are important for pausing and prey killing. Overall, this is an interesting manuscript; however, it remains unclear what the actual function of the identified Kil proteins are.

The reviewer raised an important point because it is correct that the previous data did not formally establish that a Tad-like machinery is recruited at the prey contact site. Addressing this point was challenging because it required to either demonstrate direct interactions between KilD and Tad structural components or show that predicted Tad core components also localize dynamically upon contact with the prey. This later possibility nevertheless required to obtain functional fluorescent protein fusions, which are typically difficult to obtain for membrane proteins. Below we describe which strategies we chose to address the reviewer’s comments.

Weaknesses include:1) The lack of genetic complementation experiments. Thus, it is unclear precisely which of the Kil proteins are important for predation.

This question is especially relevant for the cluster 2 genes, given that its functional association with the cluster 1 genes were only provided genetically in the previous version. In this cluster, we have chosen to only delete the genes annotated as Tad-like proteins, namely the two IM platform proteins CpaA and CpaG, the outer membrane accessory protein CpaB and three predicted pilin homologs. We did not attempt complementing the pili deletions given that they all show at best intermediate phenotypes when individually deleted and that a triple deletion is needed to obtain a kil-null phenotype. This strongly argues against polar effects in the pilin deletion mutants. However, we agree that it was important to show that the mutation in the Cpa homologs were not polar, demonstrating the critical function of these genes. In this version for coherence, we chose to complement core Tad components encoded in cluster 1 and 2, the secretin (cluster 1), the ATPase (cluster 1) and the two IM platform proteins (cluster 2). These complementations are now provided as proof that these genes are all essential for predation in a new Figure 5a and S4b.

2) The Kil proteins are encoded in two gene clusters. The evidence that these proteins make up a Tad pilus system is based on homology and that mutations in both clusters result in reduced predation. No evidence is presented that proteins encoded by these two clusters interact to form a Tad pilus machine.3) The authors localize the Kil system using an NG-KilD fusion; however, there is no evidence that KilD, which is a FHA domain-containing protein, associates with the Tad pilus machinery. In fact, KilD makes clusters independently of all other Kil proteins tested suggesting that these clusters may not report on Kil assembly and activity. An equally plausible scenario is that *Myxococcus/E. coli* contacts result in activation of KilD leading to the formation of foci. These foci then signal assembly of the Kil system somewhere in a cell (or maybe not). Therefore, it is not clear where and if this machinery localizes during prey contact.

These two points are related so we answer them jointly.

Showing direct interactions between Tad proteins is challenging and in fact, there is currently very little interaction data for these machineries, contrarily to Type-IV pili and Type-2 secretion systems.

For this reason, we chose a localization approach reasoning that localization of Tad core components in contact with *E. coli* would show that the system is assembled at the prey contact site. We now present data showing that both KilF (The ATPase encoded by cluster 1) and KilG (the CpaG homolog) both form clusters at the prey contact site similar to KilD. Since these proteins are predicted to form complementary parts of the Tad machinery and are encoded by cluster 1 and 2, we believe that these results demonstrate dynamic assembly of the machinery at the prey contact site.

4) I did not find a description of how the mutagenesis was done. Please include a description of how the mutagenesis was done, how many mutants were screened, and in which loci the mutations (transposon insertions?) occurred. Was the screen saturated?

We did not perform an extended genetic screen to find the *kil* genes. We tested a number of selected mutants in predicted membrane complexes, including all A-motility genes, possible orthologs of the Caulobacter Cdcz system, a possible CDI system, T6SS genes and decarboxylase genes and the *kil* cluster 1 and 2 genes. Mutations in the *kil* cluster 1 and 2 genes were the only ones to show a killing defect so we followed up. We do not mention all the tested genes, given that they were not investigated in depth and rapidly discarded as negative candidates.

We nevertheless clarified the text to avoid any confusion.

5) Throughout the manuscript, the authors need to tone down their conclusions and stick to what they actually show. It is also important that the authors present their results in the context of what is already known about contact-dependent killing in M. xanthus.

We believe that this comment was mostly an objection to our inference that our data previously showed assembly of the Tad pilus at the contact site. The new data strongly reinforces this view. We nevertheless carefully rewrote the manuscript making sure that the conclusions are indeed in line with the data.

6) Line 20: delete "perhaps". It is well established from the work in ref. 7 that killing can occur in a contact-dependent manner.

Corrected.

7) Line 22-23: There is no evidence in the manuscript that killing is driven by the Kil system. The evidence presented shows that the Kil system is important for killing.

True and corrected.

8) Line 24: Strictly speaking the authors have shown that one Kil protein assembles at the contact site.

Correct. We now show that at least three proteins are assembled at the contact site.

9) Line 66 and 69: It has been known for more than 20 years that type IV pili and exopolysaccharide are important for S-motility. Please include the relevant references.

True and corrected.

10) Line 90-91 and Figure 2A: It is difficult/impossible to see the lysed cells in the upper panel in Figure 2A. Would it be possible to label the individual prey cells with a number or in different colors to be able to follow cells over time?

We labeled cell with numbers to clarify this point. While it can be difficult to assign the lysed cells in the Figure 2a panel, this is facilitated by the accompanying Video 3, which contains all the frames and associated segmentation.

11) Line 91: Delete "remarkably" because this has been seen in ref. 7.

Corrected.

12) Line 103-105: This detailed description fits to the "lower" *E. coli* cell but not to the "upper" *E. coli* cell in Figure 2D. Is that important? How many cells were analyzed? Is the cell in Figure 2E representative?

In fact, it fits to both. Bi-directional signal decay from the contact site is also seen for the upper cell (Shown here, for the reviewer) but is only shown as a kymograph representation for the lower cell. We have performed this analysis for a total of 10 cells.

13) Line 124: Because the killing mechanism is not identified, I suggest to rewrite to emphasize that you looked for genes important for killing.

Rewritten.

14) Line 124-131: As described, my understanding is that M. xanthus and *E. coli* cells are mixed in suspension culture and with shaking. From the data in Figure 1 and 2, one gets the impression that predator and prey have to be in close contact for several minutes before the *E. coli* prey lyses. How would this happen in suspension culture with shaking?

This is in important question also asked by Reviewer 3. To address this in more details, we revisited the requirement of TFPs and the Agl/Glt system for killing in liquid. We found that TFPs are required, but not the Agl-Glt system, which here is the opposite of the requirements that we see on surfaces. We also performed a crystal violet assay, which revealed that aggregation of predatory and prey cells is induced by prey cells in liquid, likely a TFP and EPS dependent effect. Thus, the A-motility and S-motility complexes are differentially required depending on the conditions, solid or liquid, but none of them are absolutely required for killing. The Kil system is on the contrary required in both conditions, and thus the functions of the A- and S-motility complexes are indirect.

15) Figure 4A, B: What precisely is shown in these two figures? And how are the figures different?

The figures are two representations of the same data meant to show the dynamic of three clusters, precisely in a kymograph in panel B. We removed the kymograph as it did not provide much important information.

16) Line 204-205: How do the experiments described lead to the conclusion that the "…unambiguously that cluster formation reflects contact-dependent killing"? Is it not more correct to say that your data support that cluster formation correlates with killing?

Formally true, hence corrected.

17) Figure 4C and accompanying text: In how many cells was the reported pattern observed?

This correlation as 100% for n=10 tested cells.

18) Line 213: There is not a 2-reduction is foci formation according to figure 4D.

We removed the statement: no statistical difference is observed.

19) Figure 5D: What do the red and green lines indicate?

This graph shows traces of tracked cells for which we measured changes in cell length over time to monitor growth at the single cell levels. The two traces show that WT cells grow and divide (red line), while the kill mutant does not and in fact tends to shrink (green line). The statistical compilation of this graph is provided in Figure 5e.

Reviewer #2:This works has clear novelty and describes aspects on how Myxococcus xanthus can kill other bacterial cells.It starts by demonstrating that A-motility (Agl-Glt system) is needed to invade adjacent colony. These A-motile cells are able to kill a prey (*E. coli*) likely by making holes in the peptidoglycan layer, but the killing does not directly involve the A-motility system.Based on this, the authors did use an elegant approach to identify mutants that can invade but cannot kill. Reported hits lie within two gene clusters encoding Tad proteins, which in other bacteria such as *Pseudomonas aeruginosa* are involved in the assembly of a Tad pilus which promotes bacterial attachment. The cluster 1 encodes the prepilin peptidase, the secretin and the ATPase, while cluster 2 encodes the inner membrane platform, major and minor pilins. Both clusters encode additional genes of unknown function. It is then shown that this Tad-like system, called Kil system, can trigger target cell lysis probably via the recruitment of other systems allowing delivery of toxic elements into prey cells. Despite not having data supporting what could contribute to the toxicity, it is shown that the Kil system is actually assembling at the site of contact with the prey cell. Finally, the authors also showed that the Myxococcus Kil system allows killing of a wide range of bacteria which are not necessarily phylogenetically-related.In conclusion, this work brought novel and original concepts, some of which would definitively deserve further investigation in subsequent studies.This is a paper that I previously reviewed for another journal.Most of the queries I had have been addressed experimentally or discussed in the text. This includes further repeats on a number of experiments which are now providing statistically significant data supporting solid conclusions, including revisiting the killing of Bacillus.There are still pending questions such as what is the effective mechanism which leads to killing upon Tad-dependent contact, is there a Tad-like pilus assembled or why is the KilA prepilin peptidase not essential for killing. I would agree that these are likely beyond the scope of the present study.

We thank the reviewer for the positive assessment.

In this revised manuscript, we now provide data that we believe strongly reinforce the view that a Tad machinery assembled form the cluster 1 and 2 genes is indeed assembled at the contact sites.

As discussed in the text, it is likely that the prepilin peptidase is dispensable due to the action of another one, these enzymes being known as promiscuous. We could unfortunately not test this hypothesis because the obvious candidate the Tfp prepilin peptidase PilD seems to be essential in *Myxococcus.* (We recently confirmed this observation in an as yet unpublished TnSeq study observing that the *pilD* gene is the only *pil* gene where no Tn insertion is recovered).

Now writing "mysterious mechanism" in the abstract (line 20) might give the feeling that the story is incomplete, which should be avoided.Another yet intriguing question would be whether some of the genes of unknown function encoded within each of the tad-like clusters are involved in killing.

This is indeed a crucial question and we are now testing this possibility by making systematic gene deletions in all cluster 2 genes. However, at the current stage we don’t have evidence that toxic system is encoded by these genes, a search that will likely take another full research project.

Reviewer #3:The authors set out to identify factors involved in bacterial predation and have identified a set of genes that are homologous to secretion systems. The study focuses on M. xanthus as the predator with *E. coli* as the prey source. The authors demonstrate that M. xanthus A-Motility (focal adhesion mediated motility) is required for efficient penetration of *E. col*i under the conditions of their assays. The authors identified two genes that encode proteins thought to assemble into a type IV filament-like machine, designated "Kil". This system inhibits motility of prey cells and stimulates lysis of those cells. They show that the Kil apparatus assembles near contact sites between predator and prey cells using protein-fusion constructs. However, there are limitations based on experimental design that diminish support for the stated conclusions. The assay used to identify genes of interest was conducted in aqueous media where the motility system of interest is not required. Furthermore, the microscopic techniques used here do not allow for the visualization of pores or precise localization of machinery thought to be involved in the mechanism for delivery of toxins from predator to prey cells.I enjoyed reading this story but have several points that I would like to see addressed in a modified manuscript.

We thank the reviewer for this enthusiastic response.

ResultsLine 72-82.The authors explore prey invasion by M. xanthus wild type, pilA and aglQ mutants. The aglQ strain did not efficiently penetrate the prey while the pilA did. However, Figure 1 shows the wild type and the aglQ strain. I would like to see the pilA strain for comparison. Similarly, the Figure legends for Video 1 and 2 are the same; it is not clear which cells were labeled with mCherry.

We have now included the *pilA* strain as an inset, showing that A+Scells are indeed proficient at prey invasion. The legends to Video 1 and 2 are now clarified.

Line 91/92.Similar results have been reported widely in the field, most recently by Zhang et al., 2020, and those efforts should be acknowledged. The statement that A-motile cells carry a prey toxic activity suggests that cells utilizing different motility systems might utilize different toxins for predation. Is that correct?

This remark converges with reviewer 1 and we have made it more clear in the text that contact-dependent killing had been reported previously (although we stress that we had reported this work previously, emphasizing that our results were consistent with killing by plasmolysis as reported in the. Zhang et al. manuscript).

Our previous statement that “A-motile cells carry a prey toxic activity” was misleading. Addressing the killing mechanism in liquid cultures, we know show that the S-motility complex is required in these conditions. Thus S-motile cells are also likely toxic, which is also dependent on the *kil* system. In conclusion, on surfaces A-motility facilitate contact, in liquid TFPs facilitate contact and in both conditions the Kil system is required.

In Video 3 many prey cells are in contact with the predator, yet just only a small set of prey cells seemed to be lysed. This should be discussed by the authors.

It is true that not all contacts lead to prey lysis, which we show further in single cell assays. In fact 20-30% of the observed contacts are lytic (Figure 3D).

Figure 2d and Line 538: "Holes in the PG-labelling are observed at the contact site" seems to be too strong. The figure I have here does not provide enough high resolution to conclude that "holes" are visible.

This is formally correct as indeed the resolution by light microscopy cannot identify the original hole but, as shown by the kymograph, only detect PG degradation (as observed by the loss of fluorescent signal) propagating on either side of an original breach following contact. This would occur if, for example, PG hydrolases were secreted at the contact side and diffused locally. We corrected the text and only refer to local PG degradation to make this clear.

Line 124-134.The authors use a liquid predation assay to monitor lysis of *E. coli*. Upon lysis β-galactosidase is released into the media that contains CPRG substrate that is converted into a dark red substance. This is an effective assay. However, A-motility as reported to date requires a solid surface that is not present in this type of assay. Thus, the results as reported are done in an A-motility independent manner, while is suggested otherwise throughout the manuscript. Also the data for all strains tested in parallel should be shown (WT vs aglQ vs pilA). Importantly, by definition, predation includes consumption and growth of prey. Thus, colony forming units should be reported in the presence and absence of each predator strain and for the prey.

We thank the reviewer for this point which we clarified extensively in this version.

It was indeed important to clarify that killing is not a property of the Amotile cells exclusively.

For the liquid predation assay, as requested, we tested the WT, *aglQ* and *pilA* mutants, alone and in combination to show that TPFs are required in this system (see also our response to reviewer 1 on this topic). We also performed crystal violet assays which revealed that predator-prey biofilms are formed in liquid cultures. These experiments provided in Figure 2 -S8 clarify the individual functions of the A- and S-motility genes which depend on the conditions and are not themselves absolutely required for killing.

We have now added colony forming unit counts for all the tested prey and thus now predation is quantified in multiple ways:

i) Colony plate assays and CFU counts of the prey.

i) A FACS-based assay demonstrating that the *kil* genes are absolutely required for killing *E. coli* cells.

Despite several attempts we have not been able to obtain robust growth curves of the predator by CFU counting. In liquid, this difficulty is likely due to the fact it is challenging to obtain CFUs from the biofilm that is formed. On solid, it has also proven challenging to obtain CFUs during the predation cycle, a difficulty with Myxococcus cells being their auto-aggregation properties and motility.

But we do provide evidence that the Kil system is required for growth on preys by single cell tracking shown in Figure 5. This was actually a very challenging assays because it required single cell tracking overs several hours in prey fields.

All together the data shows that the *kil* genes are required for predation, meaning as the reviewer rightly points out: killing and growth.

Line 134/135.The authors use the liquid assay to screen for mutations in the predicted cell-envelope complexes, in which contact-dependent killing is abolished. However, I could not find an adequate description in the Results or the Materials and methods on how this screen was conducted. Additionally, the two location of the two mutations identified should be stated.

Reviewer 1 also raises this point, probably because by mentioning a genetic screen, we inferred that a large number of mutants (ie transposon mutants). We did not perform an extended genetic screen to find the kil genes. We tested a number of selected mutants in predicted membrane complexes, including all A-motility genes, possible orthologs of the Caulobacter Cdcz system, a possible CDI system, T6SS genes and decarboxylase genes and the *kil* cluster 1 and 2 genes. Mutations in the cluster 1 and 2 genes were the only ones to show a killing defect so we followed up. We do not mention all the tested genes, given that they were not investigated in depth and rapidly discarded as negative candidates.

We nevertheless clarified the text to avoid any confusion.

Figure 3A shows the predicted model of the Tad-like apparatus. It is not clear that this apparatus is actually made.

This comment was also raised by reviewer 1 (see our detailed answers to this reviewer).

In addition to our genetic studies, we now show that the predicted ATPase and IM Platform protein also localize at the prey contact site, strongly suggesting that a Tad-like apparatus is made.

Figure 6a, Line 620-622.The authors state that the ∆kilACF cells move over the top of the *B. subtilis* and S. typhimurium colonies without lysis. It appears to me that lysis is occuring. Again, predation (killing, consumption and growth) should be quantified by colony forming units.

The data presented was poorly quantitative. We now provide extensive CFU counts for all tested species over a time course. The data clearly shows that Myxococcus kills all tested prey in a kil-dependent manner, with the exception of *Pseudomonas aeruginosa*.

Line 317/318.Epistasis analysis is typically required to establish genetic elements within a pathway. Additional biochemical evidence could be used to support the order of events and protein interactions. That evidence is not provided here.

This is correct, the results suggest that a sensory system functions to promote Tad assembly at the prey contact site, but its exact composition is not known, and it is not clear whether KilD is part of this pathway. We are currently testing several proteins containing FHA domains in epistasis experiments, the presentation of which we believe is beyond the scope of this study. We modified the text and removed the misleading statement.

Is M. xanthus resistant to its own Kil system?

This is certainly a possibility. We are currently searching Myxococcus mutants that are not viable in the presence of a functional kil system (by Tnseq) to test this hypothesis.

[Editors’ note: what follows is the authors’ response to the second round of review.]

Essential revisions:Reviewer #2:In this study, the authors provide further evidence that predation by Myxococcus xanthus likely occurs primarily via a contact-dependent mechanism and identify one key player. They show that the motility/adhesion requirements for killing are dependent on the environment: type IV pili-mediated twitching group motility in liquid medium vs. solitary adventurous (A-type) propulsion on solid surfaces. By testing mutants in a short list of candidates (putative envelope-associated complexes), the authors identify two loci necessary for contact-dependent predation in both liquid and solid media. Primarily based on time-lapse microscopy of selected mutants, the manuscript presents sufficient evidence to support the central hypothesis that both loci are likely part of the same Tad-like pilus system (kil), and that this kil locus is involved in attachment and killing of prey cells. The exact role of kil in the process remains to be determined and it is likely that other yet unidentified factors are also necessary.The authors are for the most part cautious to not draw conclusions beyond what is supported by the data and acknowledge the remaining gaps in knowledge. However, most of the conclusions of the study hinge on the analysis of fluorescent protein foci in time-lapse videos. While the authors provide quantification of several fields, most of the figures and videos only show one or maximum two cells displaying the behavior under study. Moreover, there is no "negative control" for the videos of KilD/F/G and AglZ (A-motility) contacts, to rule out that it's not just random transient aggregation by these labelled proteins (or that contact with other cells causes temporary physical changes in those membrane regions creating some sort of artefactual foci).

We understand the comment but it is difficult to show more than one cell in the figures and show the clusters and their dynamics properly. Showing a field of cells would require to blow up of the panels, to introduce many arrow-heads pointing at clusters in various cells at distinct time points. Because the supplemental Videos are meant to show the entire time sequence corresponding to the selected cells in the figures, it would therefore not be practical to provide supplemental Videos that cover larger fields.

Additional Videos covering bigger fields could also be added (in fact, we have tried) but for killing events to be seen sufficiently well resolved, we still have to zoom on cells of interest. Thus, these Videos still show a limited number of events.

In fact, the figures showing cells should be considered as striking examples, but in-depth statistics are already provided in counts and we have thoroughly quantified the dynamics of the Neon-Green KilD fusion clusters, showing how they assemble upon contact, correlate with pauses and lysis over hundreds of cells in up to five independent replicates. This, we believe, is the real rigorous part of the analysis and it allows us to conclude that the clusters only form in contact with prey cells, their formation correlates with cells pausing and lysis. Last, the clusters localizes where PG degradation is observed. Taken together these arguments rule out unspecific aggregation and artefactual foci.

Nevertheless, the reviewer suggested that *Pseudomonas* could be an interesting control as the *Myxococcus* cells do not appear to pause and kill *Pseudomonas* cells by contact. Indeed, analyzing 150 cell contacts from two independent experiments, we could not observe cluster formation upon *Myxococcus-Pseudomonas* contacts, we have added this observation to reinforce our conclusions.

Another weakness is that I don't think the Tad phylogeny in Figure 7 supports the strong claim that "the kil genes evolved in predatory bacteria" (l. 298) or that the Tad homologs in predatory bacteria (Myxococcales, Bdellovibrionales and Bradymonadales) are a uniquely distinct clade of kil genes relative to the more distant tad homologs in other bacteria (my interpretation of l. 303-306). The Myxococcales, Bdellovibrionales and Bradymonadales tad clades are not monophyletic (as a group) as the authors seem to imply, so "suggesting a similar function" (l. 306) that's unique to these classes seems too speculative. Also, since the predatory bacteria are all δ-proteobacteria (and hence more closely related amongst them than the other clades on the phylogeny), wouldn't it be expected that their pili genes would be more similar, regardless or function? I therefore think these conclusions should be toned down. It would also be interesting for the authors to discuss if there are any predatory bacteria (or non-predatory δ-proteobacteria) where kil/tad homologues are absent.

We agree with this comment and toned down our conclusions.

The last point is interesting but only a handful of bacteria have been described as predatory, in the δ proteobacteria, but also outside of the δ branch. In addition, some bacteria are sometimes described as predatory (ie *Streptomyces*) because they kill other bacteria but they may not in fact use them as food source. In the δ proteobacteria, arguably where most predation studies have been performed, it is also not clear which are predatory and which are not. Thus, we unfortunately cannot make simple correlations between the presence of kil/Tad homologs and bacterial predation more broadly.

This section now reads:

“The kil genes are present in other predatory δ-proteobacteria. […] Prey cell penetration follows from the ability of the predatory cell to drill a hole into the prey PG at the attachment site^32^. Although this remains to be proven directly, genetic evidence has shown that the *Bdellovibrio* Kil homologs are important for prey invasion and attachment^33,34^”

– Timestamps in the videos would be extremely helpful.

We added timestamps to the Videos.

– For a reader unfamiliar with the field, the introduction does not provide enough context to understand the nuances about what predation is and how it distinguishes from other types of antagonism. One short explanatory sentence at the beginning would be very helpful. In particular, the findings from reference 7 should be explained a little bit more.

We added these precisions and wrote lines 47-56:

“In addition, *Myxococcus* cells have also been observed to kill prey cells upon contact, an intriguing process during which single motile *Myxococcus* cells were observed to stop and induce *E. coli* plasmolysis within a few minutes^7^. *Myxococcus* pauses were more frequent with live *E. coli* cells implying prey detection but the mechanism at work and the exact relevance of prey-contact killing for predation remain unclear. Potential contact-dependent mechanisms have been described in *Myxococcus*, including Type VI secretion^8^ and Outer Membrane Exchange (OME^9^). However, while these processes have been implicated in Kin recognition and homeostatic regulations within *Myxococcus* colonies^8,9^, they remain to be clearly implicated in predation. In this study, we analyzed the importance of motility and contact-dependent killing in the *Myxococcus* predation cycle”.

– At the beginning of the discussion (l. 322-325), the authors should acknowledge contact-dependent plasmolysis has been previously described (ref. 7).

We added this reference.

– I think it is very important to show videos of many fields/cells instead of just one representative cell, to show that the videos were not cherry-picked. Moreover, for the videos of KilD/F/G and AglZ (A-motility) contacts, I think it is important to show a negative control. Since the authors mention that a deletion in other kil components does not preclude foci formation (and therefore the mutant is not a control), maybe a good control are the *P. aeruginosa* videos?

See our response to the public reviews. We added the *Pseudomonas* control as suggested which is now apparent line 298-300:

*“Myxococcus* cells were however unable to form lethal clusters when mixed with *Pseudomonas* cells (tested for n=150 contacts, Video 14), suggesting that the Kil system does not assemble and thus some bacteria can evade the prey recognition mechanism.”

– I understand that it may be technically complex to do time-lapse imaging of differently labeled kil components (e.g. KilF-GPF and KilG-mCherry). However, I think showing co-localization of the foci for the two components in a single time point (which would allow longer exposure times and large numbers of cells analyzed simultaneously) would provide more compelling evidence that both proteins are part of the same complex. Similarly, it would be interesting to see if KilD/F foci colocalize with AglZ.

Co-localization studies could be interesting, but they are technically challenging to set up because it is frequent that combining dual fusions lead to loss of fusion due to the combination. We are now attempting to develop these tools, but they require further work to be used robustly and statistically. Dual fusions would be interesting to test a possible hierarchy in recruitment and timing, but it would not be more decisive than single fusions to show that the proteins are part of a complex. FRET studies could show this, but they are well beyond the scope of this paper. The fact that each tested protein accumulates at the prey contact site prior to lysis and that they are all required for lysis and predation is already strong indication that they function together.

It is difficult to observe both AglZ and KilD/F foci in the same cells for the same reasons (even more so, given the difficulty in tracking focal adhesion complexes). But we would not expect to see these proteins co-localize because AglZ complexes become dispersed when the cells pause (ie Figure 2b) and thus when the Kil clusters form.

– Do any of the other genes in the two loci have sequence signatures of toxins? (For instamce, to the HMMs compiled in Zhang D, de Souza RF, Anantharaman V, Iyer LM, Aravind L. Polymorphic toxin systems: Comprehensive characterization of trafficking modes, processing, mechanisms of action, immunity and ecology using comparative genomics. Biol Direct. 2012;7:18.)

Thanks for the suggestion, but unfortunately that is not the case. In fact, we have asked Aravind L to look at these genes and he did not find any obvious signatures.

– L. 232 there is a typo; should read "degradation was detected".

Corrected.